# Sub-nanoliter metabolomics via mass spectrometry to characterize volume-limited samples

Yafeng Li[1], Marcos Bouza[1], Changsheng Wu[2], Hengyu Guo [2], Danning Huang[1], Gilad Doron [3], Johnna S. Temenoff[3,4], Arlene A. Stecenko[5], Zhong Lin Wang [2,6] & Facundo M. Fernández [1,4✉]

The human metabolome provides a window into the mechanisms and biomarkers of various diseases. However, because of limited availability, many sample types are still difficult to study by metabolomic analyses. Here, we present a mass spectrometry (MS)-based metabolomics strategy that only consumes sub-nanoliter sample volumes. The approach consists of combining a customized metabolomics workflow with a pulsed MS ion generation method, known as triboelectric nanogenerator inductive nanoelectrospray ionization (TENGi nanoESI) MS. Samples tested with this approach include exhaled breath condensate collected from cystic fibrosis patients as well as in vitro-cultured human mesenchymal stromal cells. Both test samples are only available in minimum amounts. Experiments show that picoliter-volume spray pulses suffice to generate high-quality spectral fingerprints, which increase the information density produced per unit sample volume. This TENGi nanoESI strategy has the potential to fill in the gap in metabolomics where liquid chromatography-MS-based analyses cannot be applied. Our method opens up avenues for future investigations into understanding metabolic changes caused by diseases or external stimuli.

[1] School of Chemistry and Biochemistry, Georgia Institute of Technology, Atlanta, GA 30332, USA. [2] School of Materials Science and Engineering, Georgia Institute of Technology, Atlanta, GA 30332, USA. [3] W.H. Coulter Department of Biomedical Engineering, Georgia Institute of Technology and Emory University, Atlanta, GA 30332, USA. [4] Petit Institute for Bioengineering and Bioscience, Georgia Institute of Technology, Atlanta, GA 30332, USA. [5] Emory + Children's Center for Cystic Fibrosis and Airways Disease Research and Department of Pediatrics, Emory University School of Medicine and Children's Healthcare of Atlanta, Atlanta, GA 30322, USA. [6] Beijing Institute of Nanoenergy and Nanosystems, Chinese Academy of Sciences, Beijing 100083, China. ✉email: facundo.fernandez@chemistry.gatech.edu

Metabolomics is the field of analysis that probes the complement of small molecules that is more closely related to the phenotype of biological systems. When applied to human clinical studies, metabolomics plays a vital role in understanding disease states, discovering biomarkers, and monitoring therapy efficacy[1,2]. Mass spectrometry (MS), typically hyphenated with gas or liquid chromatography (GC-MS and LC-MS, respectively), is one of the central tools in metabolomics[3]. Blood and urine are the most common biofluids studied by these analytical methods. Although state-of-the-art mass spectrometers can reach very low detection limits, the sample introduction techniques used in MS usually require certain volume of sample to work with. Therefore, many types of biospecimens remain beyond the reach of current metabolomics platforms for difficult-to-obtain biospecimens. Examples include tissue-derived cells such as hematopoietic stem cells in bone marrow[4], tears, sweat, exhaled breath condensate (EBC), infant or child-derived tissues[5], samples from longitudinal animal studies with frequent sampling, and exosomes[6]. Sample-size limitations force the exclusion of many specimens from studies, especially if metabolomics assays are combined with other −omics, such as lipidomics, transcriptomics, and proteomics. Removing sample choices because of quantity limitations effectively decreases cohort sizes and reduces the statistical significance of many potential metabolite markers. New technologies that minimize sample-size requirements for metabolomics while simultaneously increasing MS sensitivity are needed to overcome challenges in −omics studies.

An alternative approach to GC-MS or LC-MS metabolomics is direct infusion (DI) MS via electrospray ionization (ESI). The major advantage of DI MS is its high sample throughput and simplicity. Automated, chip-based nanoESI DI MS platforms have been reported to reach throughputs of >10k samples per year[7]. However, given its continuous nature, conventional ESI/nanoESI DI MS rarely achieves 100% duty cycle when coupled with pulsed mass analyzers. The result is unnecessary sample waste. Pulsed ion sources are more compatible with pulsed mass analyzers, such as Orbitrap or time-of-flight, and make the most of precious samples. In previous work, our group developed a pulsed triboelectric nanogenerator (TENG) nanoESI ion generation method[8]. The method consumes only picoliters of sample per spray pulse, while simultaneously reaching detection limits as low as 0.6 zmol.

Here we present an alternative configuration best suited for very small samples, named inductive TENG (TENGi), and its application to "tiny metabolomics". In this approach, the liquid sample is electrically charged through a contactless microelectrode surrounding the spray emitter. As the sample solution is not in direct contact with a wire electrode, constraints on requiring a minimum sample volume are effectively removed and cross-contamination between samples is avoided. Compared to traditional DI nanoESI MS, TENGi nanoESI has better signal-to-noise (S/N) ratios because of the much higher voltages generated during operation. This feature enables the generation of high-quality mass spectra with 3–5 spray pulses with only sub-nanoliters of sample.

As an example of the capabilities of TENGi MS for tiny metabolomics studies, metabolic fingerprinting of EBC collected from cystic fibrosis (CF) patients with prediabetes is conducted. CF is one of the most common life-shortening genetic diseases. It affects more than 70,000 people worldwide[9]. TENGi MS analysis is performed before and 2 h after ingestion of a glucose drink. The goal is to investigate glucose-induced changes in the airway metabolome. Studies of this kind are important in understanding the development of CF-related diabetes (CFRD), which accelerates lung disease progression in CF patients. Clinicians now recognize that CFRD has its genesis in early childhood[10], so future studies will require examining tiny EBC volumes, as collection of large volumes from infants or young children is impractical or even impossible. A second example to illustrate TENGi MS capabilities involves rare cell metabolomics of cultured mesenchymal stromal cells (MSCs), a cell type that has shown potential for treating a variety of chronic diseases[11]. Examination of metabolic changes of MSCs cultured under conditions that may impact in vitro therapeutic activity, such as aggregate culture[12], or preconditioning with interferon-γ (IFN-γ)[13], is critical for identifying attributes of cell quality. Reducing cell numbers required to perform MSC metabolomic analysis is essential for improving the manufacturing of highly therapeutic MSCs without significantly impeding production.

## Results

**TENGi mass spectrometry.** The illustration of the TENGi nanoESI ion source is shown in Fig. 1a. The type of TENG utilized here is a sliding freestanding TENG (Fig. 1b). One pair of triboelectric layers and one pair of copper film electrodes comprised the generator. The triboelectric layers were made of a nylon layer ($12 \times 12$ cm) and a fluorinated ethylene propylene (FEP) stationary layer ($24 \times 12$ cm). One output of the TENG was connected to the ion source, the other was grounded. The sample solution was loaded in a nanospray emitter. A copper electrode was used to induce spray without direct contact with the sample solution by wrapping a 10 mm-wide copper foil strip around the emitter. A photo of the TENGi nanoESI setup is shown in Fig. 1c. A custom cartridge was designed to secure the nanoESI emitter and fix the distance between the emitter tip and the MS inlet (Supplementary Fig. 1, detailed description in the "Methods" section).

The system behaved in a form equivalent to a three-capacitor circuit (Fig. 1d): the first capacitor ($C_1$) comprised the two TENG electrode/air/electrode regions, which were charged by the mechanical motion of the nylon and FEP electrodes relative to each other; the second capacitor ($C_2$) was formed by the copper electrode/glass/sample solution inside the nanoESI emitter; the third ($C_3$) comprised the sample solution/air/mass spectrometer inlet. $C_3$ could be viewed as a leaky capacitor, because when the number of accumulated charges was larger than the sample solution's ability to hold the charge, droplets were emitted and ionization begins. The TENG charging mechanism is shown in Supplementary Fig. 2, with the processes occurring at the emitter tip depicted in Fig. 1e. As a type of contactless ESI[14–16], charges are generated mainly by electrostatic field-induced molecule polarization, ionization, and charge separation. Prior to TENG actuation, the electrolytes in solution are evenly distributed to satisfy the charge balance. Once the TENG is actuated, electrostatic fields are created in $C_2$ and $C_3$ (Fig. 1e), electrospray is generated, and ionization occurs.

As TENG-powered nanoESI is a pulsed ionization method and only consumes picoliters of sample solution per pulse[8], 0.8 μL of sample was sufficient to generate thousands of signal pulses. Figure 1f is the TENGi-positive-ion mode total ion chronogram obtained from loading 0.8 μL of 1:9 v/v MeOH : $H_2O$ into the nanospray emitter. Over 1200 positive signal pulses were generated in 25 min (~0.8 Hz). Following this experiment, ~0.6 μL of unconsumed sample remained in the nanoESI emitter (Fig. 1f, right bottom panel). More signal pulses meant that more advanced MS information (e.g., $MS^2$, data-dependent acquisition) could be more readily obtained for many metabolites with only tiny sample volumes. As a practical example, 0.8 μL EBC from a healthy volunteer was loaded and subjected to TENGi nanoESI MS analysis. Positive and negative full MS spectra (Supplementary Fig. 3a, b), tandem MS (MS/MS) spectra (Supplementary

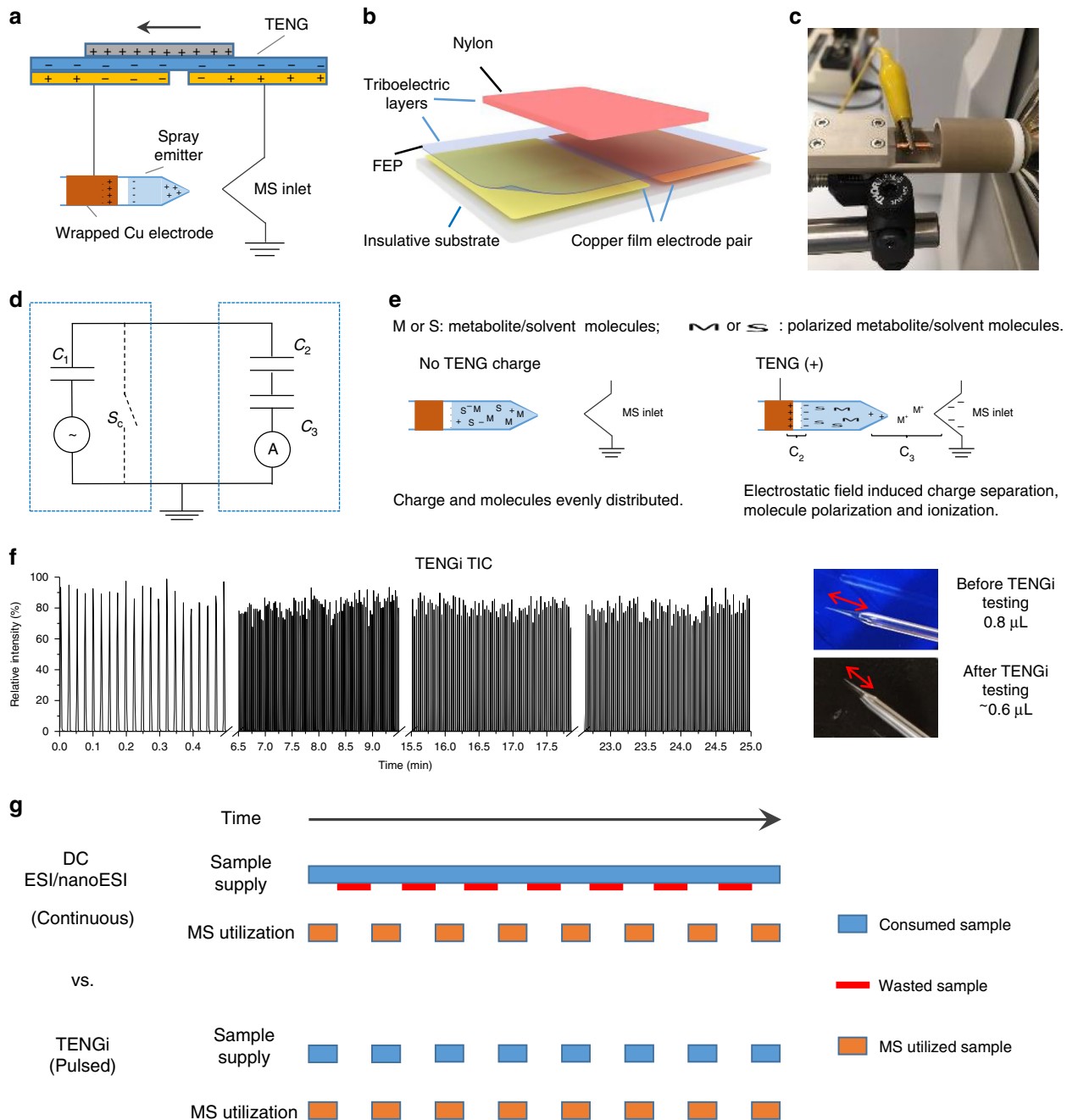

**Fig. 1 TENGi nanoESI illustration, mechanism and characteristics. a** Schematic of the TENGi nanoESI system. **b** Schematic of the SF TENG device used in this study. **c** Photo of the custom cartridge used to secure nanoESI emitters in TENGi experiments. **d** Equivalent electric circuit ($C_1$–$C_3$ depict capacitors; $S_c$ represents a short-circuit switch). **e** Charge generation mechanism in TENGi. M and S represent metabolite and solvent molecule, respectively; "+" and "−" represent positive and negative charges, respectively; $C_2$ and $C_3$, as also included in **d**, represent two capacitors formed during TENGi nESI process. **f** Total ion chronogram (TIC) produced by TENGi nanoESI of 0.8 μL of a 1 : 9 v/v mixture of MeOH : $H_2O$ together with photos of the solution contained in the emitter before and after analysis. The red arrows in the photos indicate the volume of sample solution in the nanoESI emitter. As a pulsed ionization method, the TIC appears as pulses instead of continuous signals. By sealing the larger end of the spray emitter to prevent evaporation, 0.8 μL of sample can pulse-spray for more than 25 min. **g** Illustration of sample utilization in conventional DC ESI/nanoESI and TENGi. As the mass spectrometer intermittently traps and scans ions, TENGi can be synchronized with these events so as to achieve higher sample utilization.

Fig. 3c, d), as well as ion mobility (IM)-MS data (Supplementary Fig. 3e, f) were successfully obtained. The remaining sample could be saved in a refrigerated, humidified container for more than 10 days (Supplementary Fig. 4). In cases when sample stock is extremely rare, this method could be used for sample preservation for performing metabolite annotation, or other MS experiments, at a later stage.

An additional advantage is that TENGi's pulsed characteristics are a better match to the intermittent nature of pulsed mass analyzer such as Orbitrap and time-of-flight. As illustrated in Fig. 1g, conventional DC ESI/nanoESI provides sample continuously, but only a fraction is utilized by the mass analyzer. A large fraction of the sample is wasted when the analyzer is busy scanning ions. In the case of TENGi, by synchronizing the

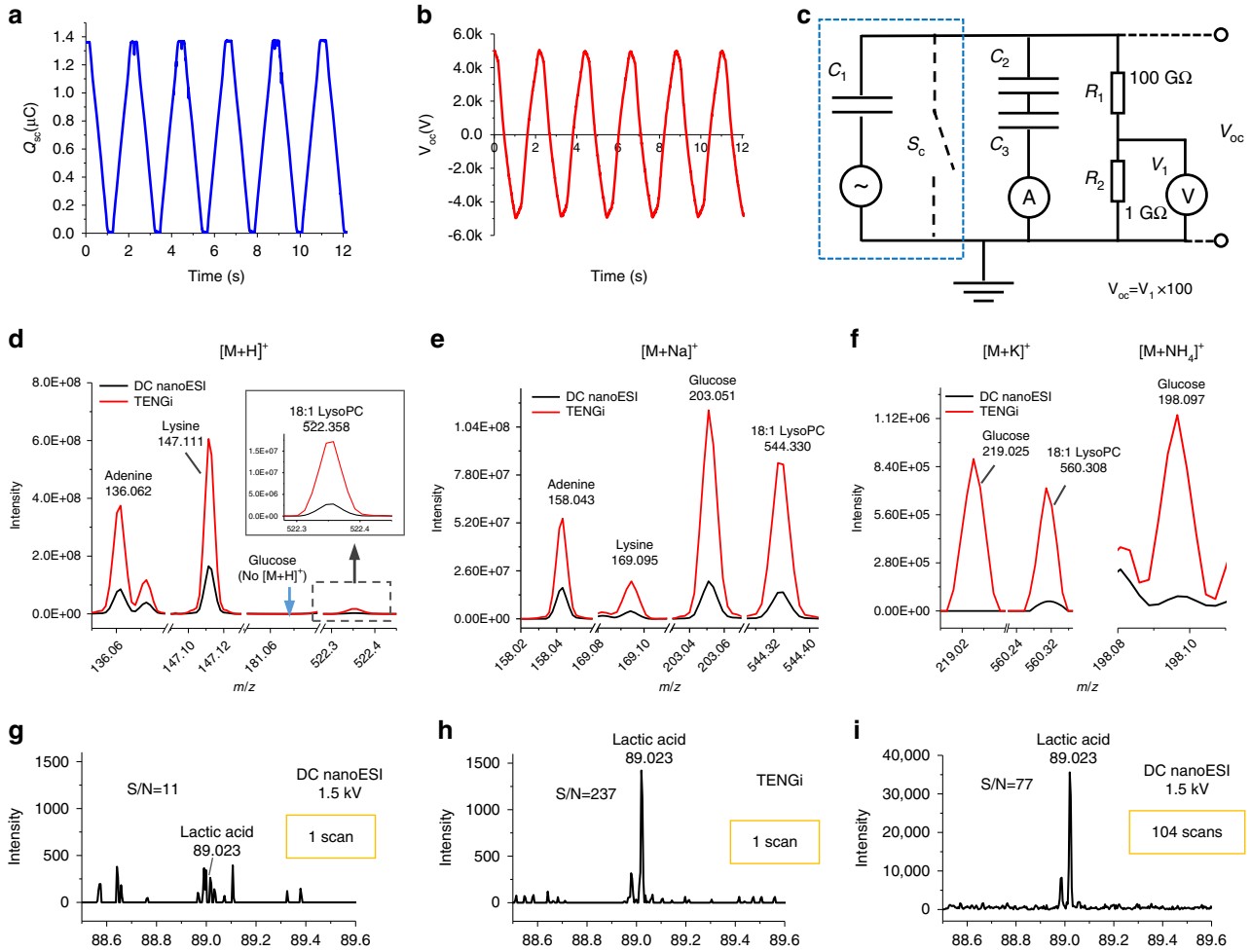

**Fig. 2 TENG device and TENGi nanoESI characterization. a** Measured short-circuit charge ($Q_{sc}$) for the TENG device used in this work. The charge generated per cycle is 1.37 μC. **b** Measured TENG output voltage ($V_{oc}$). **c** Equivalent circuit for TENG output voltage measurements, using an external load resistance of 100 GΩ to bring the voltage to the voltmeter measuring range. The blue square indicates the equivalent circuit of the TENG device. **d–f** The behavior of four typical metabolite standards (adenine, lysine, glucose, and lysoPC) with TENGi and conventional DC nanoESI. **g–i** The comparison of TENGi and DC nanoESI for the detection of lactic acid ([M-H]⁻ $m/z = 89$) in EBC of a healthy volunteer: **g** single scan conventional DC nanoESI mass spectrum, **h** single scan TENGi nanoESI mass spectrum, and **i** average mass spectrum generated from 104 nanoESI scans.

ionization and mass analyzer cycles, each spray event can be more fully utilized to generate MS information, thus largely minimizing sample waste during analysis.

Another characteristic of TENGi is its high MS sensitivity, related to the high voltage and finite charge generated. The electrical characterization of the TENG used in this work is shown in Fig. 2a, b, with additional simulations given in Supplementary Fig. 5. The maximum measured charge output was 1.37 μC (Fig. 2a). Measured TENG voltages using an external load resistor were ±5 kV (Fig. 2b, c), with the actual output voltages likely reaching much higher values due to the higher electrical resistance of the nanoelectrospray itself. COMSOL simulations were conducted to further estimate open-circuit voltages. The results showed that for the TENG device used here, voltages could potentially reach tens to hundreds of kilovolts, depending on electrode displacement distance (Supplementary Fig. 5). Such voltages are not accessible by conventional DC nanoESI, as they would lead to electrical discharging and irreversible damage to both the ESI emitter and even the mass spectrometer electronics.

The higher transient voltages reached by TENG[17,18] were associated with superior ionization performance. Four typical metabolite standards were used to test the performance of TENGi

vs. DC nanoESI (adenine, lysine, glucose, and lysoPC). In terms of ionization adduct formation, both method behaved similarly, but the ion abundances were higher with TENGi (Fig. 2d, e). TENGi was also able to detect lower abundance signals such as [glucose + K]⁺, [18 : 1 lysoPC+K]⁺, and [glucose + NH₄]⁺ whereas DC nanoESI was not (Fig. 2f and Supplementary Fig. 6). As the number of delivered charges by TENG is finite (Fig. 2a) and the generated high voltage is transient (Fig. 2b), no significant in-source fragmentation was observed (Supplementary Fig. 7). TENGi's superior performance in ionization could also be seen in the detection of lactic acid in EBC of a healthy volunteer in negative mode. Figure 2g–i show a comparison of the sensitivity improvements observed in the single scan spectra of lactic acid. Lactic acid has been detected in EBC during exacerbations[19]. Its ion abundance observed with TENGi (Fig. 2h) was five times higher than that seen by conventional DC nanoESI (Fig. 2g), with a S/N ratio gain of more than 20×. DC nanoESI required averaging more than 100 scans to obtain a mass spectrum with one-third of the S/N ratio than that of a single scan TENGi experiment (Fig. 2i). Quantification of lactic acid in EBC using TENGi is shown in Supplementary Fig. 8, with a more in-depth characterization of TENGi lactic acid pulse shape and scans-across-the-pulse given in Supplementary Fig. 9.

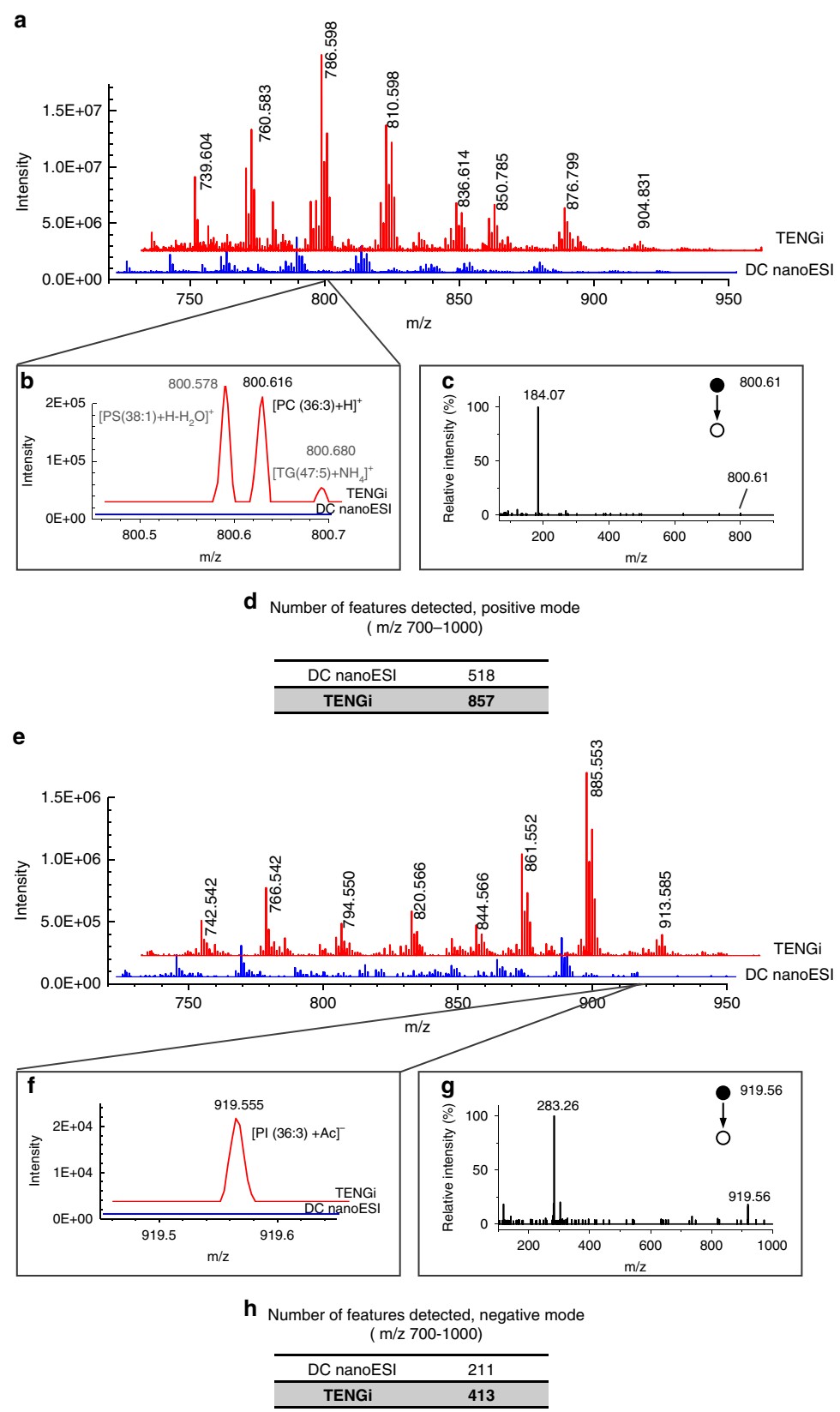

**Fig. 3 Dilute liver lipid extract (25 μg mL⁻¹) analyzed by both conventional DC nanoESI and TENGi MS. a–d** Positive-ion mode MS comparison: **a** full MS, **b** enlarged view of *m/z* 800.616 that was detected by TENGi but not by DC nanoESI, **c** TENGi MS/MS identification of the species with *m/z* 800.616; **d** number of features detected in positive-ion mode in the 700–1000 *m/z* range. **e–h** Negative-ion mode MS comparison: **e** full MS, **f** enlarged view of *m/z* 919.555 that was detected by TENGi but not by DC nanoESI. **g** MS/MS identification of the species with *m/z* 919.555; **h** number of features detected in negative-ion mode in the 700–1000 *m/z* range.

To compare the coverage of TENGi and conventional DC nanoESI for complex biological sample analysis, we used both of these techniques to analyze a dilute, 25 µg mL$^{-1}$ liver lipid extract (Fig. 3). In both positive and negative-ion full MS (Fig. 3a, e), TENGi generated much higher ion abundances than conventional DC nanoESI. Most abundant species were detected by both DC nanoESI and TENGi. However, lower abundance species, such as the ion at m/z 800.616 (Fig. 3b) and m/z 919.555 (Fig. 3f), were not detectable with DC nanoESI. TENGi was still able to detect these species, generating sufficient ion current to also carry out MS/MS experiments (Fig. 3c, g). In positive-ion mode, TENGi yielded 300 more features than DC nanoESI (Fig. 3d); in negative-ion mode, the number of features detected by TENGi was almost twice that of DC nanoESI (Fig. 3h).

**Sub-nanoliter metabolomics for CF patients.** CF begins in infancy. Monitoring metabolic phenotype changes in the EBC of CF infants or children has been identified as a promising approach for monitoring disease progression and implementing early interventions[20,21]. CFRD is one of the most frequent co-morbidities in CF with 20% of teens and 50% of adults developing diabetes. Morbidity and mortality are significantly more common in CF with diabetes compared to without the diabetes complications. Death is attributed to accelerated decline in lung function and eventual respiratory failure, not cardiovascular complications[22]. However, it is not yet well understood why metabolic abnormalities, such as those in glucose metabolism, exacerbate lung function loss. The lack of understanding is partly because of the relative inaccessibility of the affected tissues and the limited availability of representative animal models[22]. For these reasons, studying changes in the EBC metabolome as CF patients develop progressively worse abnormalities in glucose metabolism and end up with CFRD holds promise as a means for understanding CFRD pathophysiology. As it is now believed that CFRD has its genesis in early childhood, TENGi metabolomics on EBC collected from young children may enable studying the early biochemical processes associated with CFRD.

EBC is attractive as a biofluid, as it can be non-invasively collected[20]. However, it is highly diluted by exhaled water vapor, typically requiring at least 1 mL of sample, which is 20-fold concentrated to yield informative LC-MS data[19]. Teenagers and adults normally require more than 10 min of tidal breathing through tube collection devices, to obtain 1 mL of EBC. Subjects are allowed to take short breaks but this collection can be very challenging or even impossible for toddlers and infants. We aimed to address the feasibility of assessing the airway metabolic profile in young children because glucose metabolism worsens as CFRD develops. We identified CF patients (Supplementary Table 1) with early abnormalities in glucose homeostasis, as defined by having prediabetes/impaired glucose tolerance (CF IGT) detected via an oral glucose tolerance test. EBC from these CF patients were collected both prior to (fasting) and 2 h following the ingestion of a glucose drink. These two sets of EBC samples were termed Pre and Post for simplicity, with ten EBC samples per group. The overall workflow implemented for EBC TENGi metabolomics is presented in Fig. 4a. Only 50 µL of EBC was used per sample for this workflow, which was equivalent to <1 min collection.

Several important steps were implemented to ensure maximum reproducibility in TENGi spectral fingerprints. Reproducibility is critical in metabolomics because of its sensitivity to small perturbations. These steps included as follows: (1) use of commercial nanoESI emitters to ensure reproducible spray-tip diameter; (2) a custom-designed cartridge to support the emitters (Fig. 1c and Supplementary Fig. 1), maximizing spray position reproducibility; (3) a mechanical motor to standardize TENG actuation frequency and displacement distance; and (4) spiking the samples with a stable-isotope-labeled internal standard (IS) to monitor the quality of each pulse. Quality control (QC) samples were inter-dispersed with EBC samples to monitor the instrument's technical variance (Fig. 4b). A strict multi-step data processing protocol was implemented (Fig. 4c) that included removal of low abundance signals, IS normalization, filtering of blank signals, missing value filtering and imputation, etc., to guarantee the quality of the final dataset and reliable metabolomics results.

Isotopically-labeled $^{13}$C tyrosine was used as an IS and its extracted ion chronogram area was monitored throughout all TENGi experiments to ensure the highest quality data and identify any outliers. Pulses with IS integrated areas outside of the 20% expected median relative SD (RSD) were discarded (see "Methods" section). The RSD of all peak abundances in the QC samples[23] (RSD$^{QC}$) was calculated for every spectral feature detected within a given experimental run. This step was carried out for the same nanoESI emitter as well as across three different emitters. As shown in Fig. 5, RSD$^{QC}$ values for runs with the same nanoESI emitter and runs done with different emitters were 12.8% and 17.2%, respectively. These values were acceptable levels specified by US Food and Drug Administration guidelines for biomarker studies[24].

Following MS data collection and processing, the dataset was subjected to multivariate analysis to investigate glucose-induced changes in airway metabolic profiles. A partial least squares-discriminant analysis (PLS-DA) score plot comparing Pre and Post groups is shown in Fig. 6a. The plot indicates good separation of the two groups. To focus on the most important features that contributed to the separation of these two groups, only features that passed both the volcano plot criteria (p-value < 0.05, fold change (FC) > 2 or FC < 0.5, Supplementary Fig. 11a) and had a PLS-DA VIP score > 1.8 (Fig. 6b) were selected for the final discriminant feature list. Five metabolic features were the most significant ones in distinguishing the Pre and Post groups. The orthogonal PLS-DA (oPLS-DA) classification plot for a model constructed using only the 5 most important features is shown in Supplementary Fig. 11b, with a sensitivity of 89% and a specificity of 70% under cross-validation. Putative, high-resolution, and accurate mass annotations for these features are provided in Supplementary Table 2, with the experimentally-observed accurate masses and isotopic distributions shown in Supplementary Fig. 12. MS/MS information is provided in Supplementary Table 3. Box plots for these five significant metabolites are shown in Fig. 6c–g, illustrating the significant changes observed in the EBC after 2 h of the glucose challenge test. Interestingly, all five metabolites were found to increase. A heat map depicting the relative abundances of such metabolites by class is shown in Supplementary Fig. 13, indicating the level of each feature in each sample, and correlations between features.

TENGi studies on CF IGT patients revealed interesting metabolic alterations following glucose challenge. Feature #1 was tentatively identified as a glycerophosphoserine PS(39:3), by accurate mass measurements and MS/MS (Supplementary Table 3). Phosphatidylserines are membrane phospholipids normally restricted to the inner layer of the cell lipid bilayer; their externalization is one of the earliest events in programmed cell death[25]. High glucose levels can induce cell death through a free radical-mediated mechanism[26]. In patients with diabetes, high glucose levels can cause endothelial cell damage in blood vessels[27,28], aortic smooth muscle cell death[29], and neuropathy-associated cell death[30]. Increase in PS(39:3) abundance in EBC after glucose ingestion suggests that high glucose levels may also induce apoptotic events in the lung microenvironment. Feature #2 was identified by accurate mass and MS/MS as

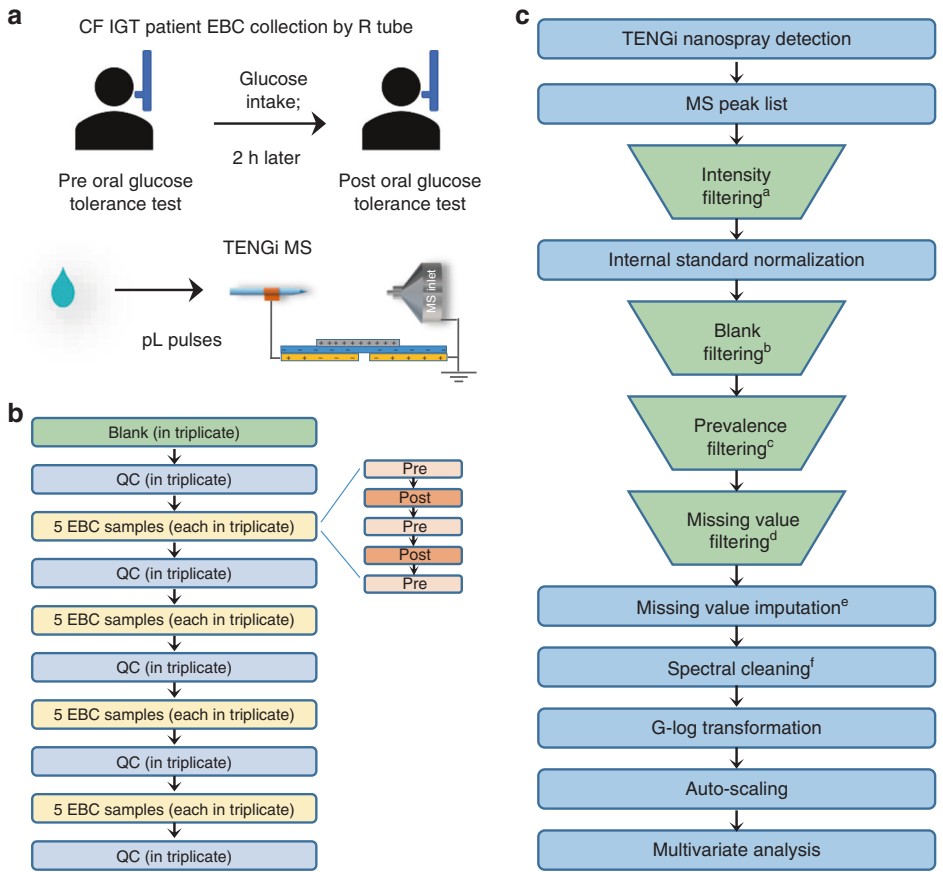

**Fig. 4 Experimental and data processing workflow for TENGi MS EBC metabolomics. a** Illustration of EBC sample collection and TENGi MS analysis; **b** TENGi MS run order, including method blanks, quality control (QC) samples, and EBC sample replicates. Pre and Post samples were run in an interleaved fashion within each sample segment, and randomized in terms of their overall order. Three replicate tests were conducted for each sample. Outliers were discarded based on the extracted ion chronogram pulse area of the spiked IS; **c** TENGi MS data processing protocol. [a]Spectral features with abundances lower than 2000 were removed from the dataset. [b]Blank filtering was applied to remove features with normalized abundances lower than 3 times that of the blank. [c]A spectral feature had to be detected in at least 80% of the EBC samples, otherwise it was removed. [d]Spectral features in the QCs with 80% missing values or not detected were filtered out and sample(s) with more than 20% missing values were deemed as outlier(s) and removed (Supplementary Fig. 10). [e]Missing values were replaced by half of the minimum positive value in the original data. [f]QC features with median relative SD (RSD$^{QC}$) > 30% were removed. The order of the four filtering processes is interchangeable.

trimetaphosphoric acid, whereas feature #3 was tentatively identified by accurate mass and partial MS/MS (Supplementary Table 3) as lactosylceramide (LacCer)(t31:1). Not surprisingly, both metabolites are related to the lung microbiome, which is altered in CF patients. Trimetaphosphoric acid has been reported as a hydrolysis product of lipopolysaccharides, which are characteristic of gram-negative bacteria[31,32]. LacCer is a glycosphingolipid that plays a vital role in the biosynthesis of asialoGM1[33,34], a known CF respiratory epithelial surface receptor for bacterial pilin[35]. As bacterial infection in CF patients is common[36–38], increases in these compounds might suggest that glucose challenge is also inducing airway bacteria metabolic changes. Feature #4 and #5 were tentatively identified by accurate mass and partial MS/MS (Supplementary Table 3) as a coenzyme A and mannose-(inositol phosphate)2-ceramide (d34:0), respectively. The roles of these metabolites in the context of this study are not yet clear. Further research with larger cohorts is warranted to further study their roles and yield clinically meaningful conclusions.

This proof-of-concept study using TENGi MS and sub-nanoliter quantities of EBC successfully demonstrated the potential capabilities of the technology for airway metabolic phenotyping. The ability to detect metabolomic changes with minimum sample amounts collected from the same subject after a few hours following a metabolic challenge opens new possibilities

for investigation where traditional approaches would typically not be applicable.

**Sub-nanoliter metabolomics on human MSCs**. Therapies based on MSCs have shown promise for treating a number of orthopedic, autoimmune, cardiovascular, and neurodegenerative diseases[39,40]. These cells have characteristics similar to stem cells and can differentiate into a number of different cell types. Metabolomics studies on stem cells could provide insights into their level of differentiation or secretory function[12,41,42], give better control of scale-up processing[43], reveal differences in donor sources[44], and offer information on their efficacy[45,46]. In addition, MSC culture conditions can significantly impact their immunomodulatory potential when transplanted into inflammatory environments[47]. Preconditioning MSCs with inflammatory molecules, such as IFN-γ, has been shown to improve their ability to reduce immune cell (T-cell) proliferation in vitro[13,48]. Therefore, understanding MSC response under inflammatory conditions is essential for developing and improving MSC-based therapies.

As MSCs are generally derived from patient donors, they are limited in supply and costly to expand to sufficient numbers for metabolomics testing. This is a common issue with many cell types, such as hematopoietic stem cells from bone marrow or

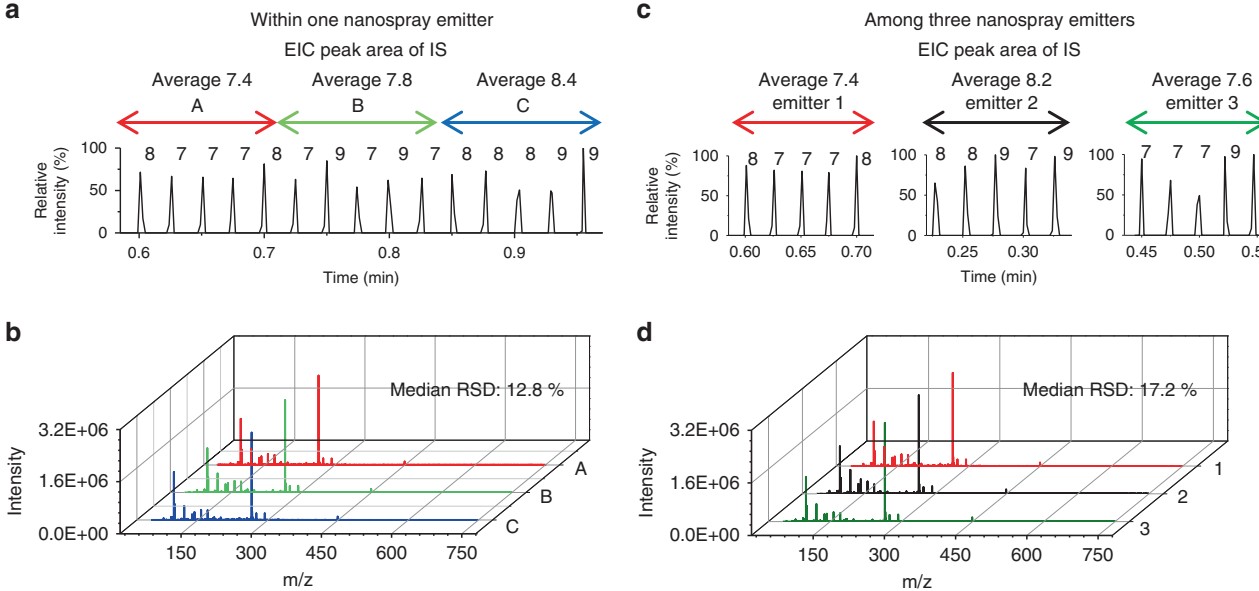

**Fig. 5 Reproducibility study of triboelectric ion source for tiny metabolomics.** TENGi MS reproducibility using the same nanoESI emitter (**a**, **b**) and between three different nanoESI emitters (**c**, **d**). In all cases, the sample sprayed was a QC sample made up with pooled EBC. **a**, **c** Extracted ion chronograms for the IS. The label displayed over each individual pulse (7, 8, 9, etc.) represents the integrated area of the IS, $^{13}$C tyrosine. Different colors represent different time regions. **b**, **d** The corresponding averaged mass spectra for the TENGi pulses shown in regions A, B, C and 1, 2, 3, respectively. The colors match those for the regions shown in **a** and **c**. The median intra and inter-emitter RSD$^{QC}$ values shown in **b** and **d** were 12.8% and 17.2%, respectively. Source data are provided as a Source Data file.

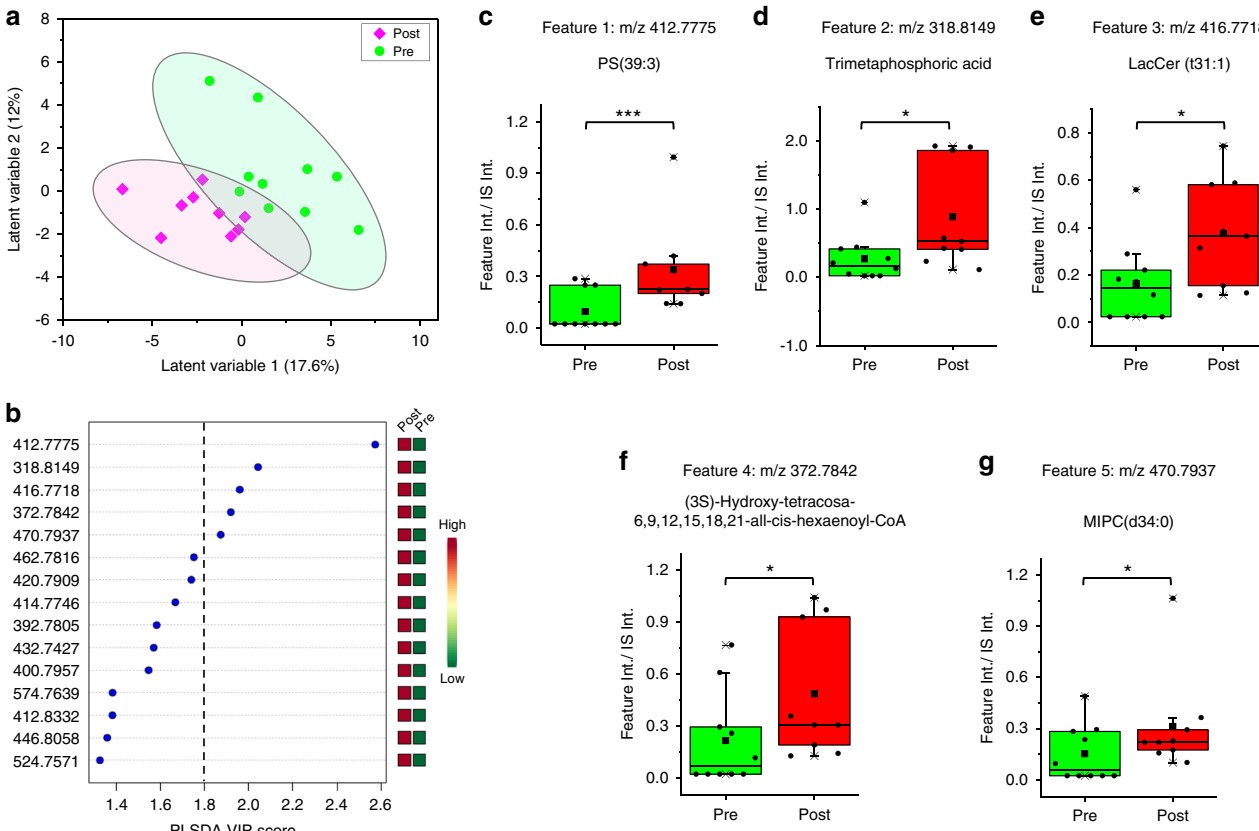

**Fig. 6 TENGi metabolomics results for the comparison of CF IGT patients pre- and post-oral glucose tolerance test. a** PLS-DA score plot using all features. **b** PLS-DA VIP score plot. The vertical dashed line indicates the VIP score > 1.8 cutoff. Five metabolic features were selected based on this cutoff. **c**–**g** Box plots of IS-normalized MS peak abundances for these five features. For the Pre group, $n = 10$ biologically independent samples were tested; for Post group, $n = 9$ biologically independent samples were tested. Differences between the two groups were evaluated using the two-tailed Student's $t$-test (\*\*\**p*-value < 0.001, \**p*-value < 0.05, detailed *p*-values are: 0.00086, 0.014, 0.019, 0.022, 0.026, respectively. These were calculated based on the final dataset after G-log transformation and autoscaling through MetaboAnalyst). Source data are provided as Source Data files. The lower, middle and upper lines in box plots **c**–**g** correspond to 25th, 50th, and 75th percentiles. The whiskers extend to the most extreme data point within 1.5 interquartile range (IQR).

tumor-derived stromal cells that are difficult to collect in large-enough numbers for metabolomics, leading to an increasing need for rare cell metabolomics methods. Li et al.[49], e.g., developed an isotope labeling nanoLC-MS method for small cell number untargeted metabolomics that has shown promise. Morrison et al.[4] developed a targeted metabolomics method to screen 50 different metabolites in haematopoietic stem cells and restricted haematopoietic progenitors.

To test TENGi capabilities for small cell number metabolomics, MSCs cultured under normal and IFN-γ-stimulated inflammatory culture conditions were studied via TENGi MS. The analysis used

only 80,000 cells per sample. The experimental workflow is illustrated in Fig. 7a, with more details on the experimental design and approach provided in Supplementary Fig. 14 and the "Methods" section. Data were processed with a workflow similar to the one used for EBC metabolomics (Fig. 4c). The resulting PLS-DA score plot comparing stimulated and unstimulated MSC groups is presented in Supplementary Fig. 15a, showing clear clustering of samples in these two groups. Corresponding loadings plot for all the features, and the top 50 features ranked by their PLS-DA VIP scores are shown in Supplementary Fig. 15b, c, respectively.

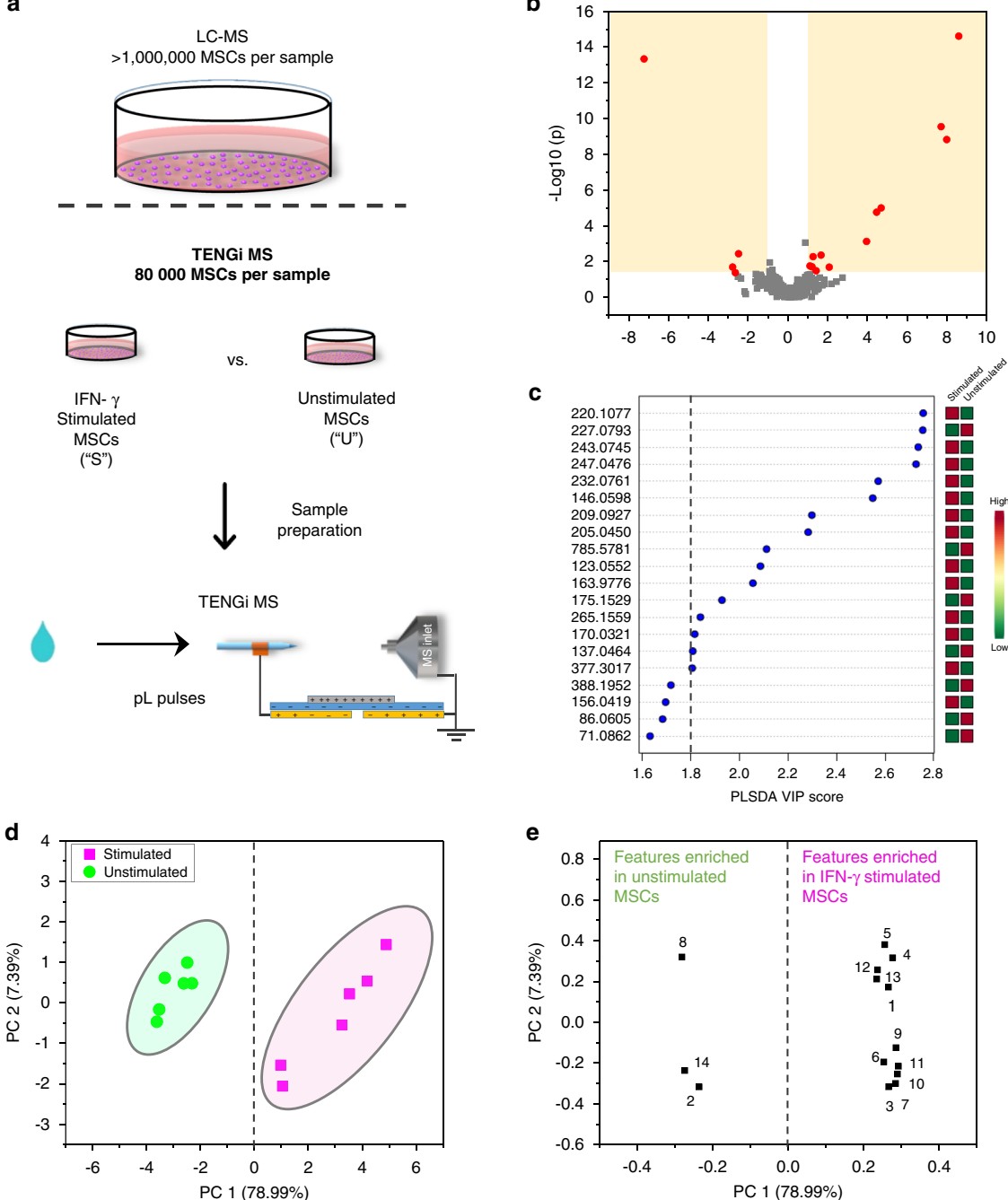

**Fig. 7 TENGi sub-nanoliter metabolomic studies of IFN-γ-stimulated MSCs. a** TENGi MS workflow; **b** volcano plot for IFN-γ stimulated (S) vs. unstimulated (U) MSCs. **c** PLS-DA VIP score plot with the top 20 features ranked by their PLS-DA VIP scores. **d** Principal component analysis score plot for IFN-γ stimulated vs. unstimulated MSCs using 14 significant features selected by volcano plot criteria and PLS-DA VIP score >1.8; **e** corresponding principal component analysis score loadings plot identifying features that discriminate stimulated vs. non-stimulated MSCs.

To focus on the most important features that separated stimulated from unstimulated cells, only features that passed both the volcano plot criteria (p-value < 0.05, FC > 2 or FC < 0.5; Fig. 7b and Supplementary Table 4) and had a PLS-DA VIP score > 1.8 (Fig. 7c) were chosen for the final feature set. The 14 features selected with these criteria were annotated using accurate, high-resolution MS and MS/MS measurements with good success (Table 1). One feature (#3) was annotated with the use of IM-MS/MS measurements, as precursor ion co-selection was observed in standard MS/MS experiments. The principal component analysis (PCA) score plot for these 14 features showed very clear clusters (Fig. 7d) and marked contributions to the principal component 1 and 2 loadings (Fig. 7e). Furthermore, oPLS-DA modeling with these 14 selected features yielded a classification accuracy of 100% under leave-one-out cross-validation (Supplementary Fig. 15d), indicating that these 14 features are highly relevant to the effects of IFN-γ challenge. Box plots for these 14 features are shown in Supplementary Fig. 17.

TENGi MS metabolomic studies on stimulated MSCs revealed that, under inflammatory conditions, eleven of the highest-scoring metabolites were enriched, while three (#2, 8 and 14) showed a decrease in abundance (Fig. 7e). Capturing these metabolic differences in MSCs cultured under different conditions is critical to efficient identification of therapeutic cell quality biomarkers[50,51]. Furthermore, development of metabolomic probes that can significantly reduce the required number of cells or samples could enable their integration into current cell manufacturing processes and help preserve the majority of produced cells for patient use[52].

Examination of FC values in Table 1 showed that hypoxanthine, a product of ATP breakdown[53], sharply decreased under IFN-γ stimulation. Oxidative stress, associated with inflammatory processes, is hypothesized to cause sustained damage to mitochondria and result in ATP depletion[54]. Feature #4, identified as taurine, was seen to significantly increase in abundance. As this metabolite is known to play an important antioxidant role and provide resistance against reactive oxygen species[55–58], its increase in abundance is likely associated with protective mechanisms triggered in MSCs upon IFN-γ stimulation. Similar trends in taurine concentration after IFN-γ MSC stimulation have been previously reported by Vunjak-Novakovic et al.[59] using GC-MS and 2 million cells per sample, 20 times more than the number we used here.

The heat map in Fig. 8a shows the distribution of the top 14 features for each MSC sample studied. Pathway analysis showed that tryptophan catabolism was altered through three different pathways (Fig. 8b and Supplementary Fig. 18) with L-kynurenine, 5-hydroxy-L-tryptophan (5-HTP), and 3-methyldioxyindole (3-MDI) being the key products. MS/MS spectra for L-tryptophan, L-kynurenine, and 5-HTP in MSC samples, and the corresponding MS/MS spectra for chemical standards are shown in Supplementary Fig. 19. The MS/MS spectrum of 3-MDI obtained via IM-MS/MS is shown in Supplementary Fig. 20. A Pearson's correlation study was conducted to identify features associated with tryptophan (m/z 227.0793). As shown in Fig. 8c, tryptophan was inversely correlated with six other species, five of which are tryptophan catabolic products shown in Fig. 8d: 5-HTP (m/z 220.1077, m/z 243.0745), L-kynurenine (m/z 209.0927, m/z 247.0476, box plot shown in Supplementary Fig. 17), and 3-MDI (m/z 146.0598). The bar chart in Fig. 6d shows that the absolute correlation coefficient for these five features were all >0.8. Overall, these results further confirmed that IFN-γ stimulation affects MSC tryptophan metabolism through three different pathways.

It is well established that when MSCs are challenged by IFN-γ, there is altered tryptophan catabolism through the kynurenine metabolic pathway[60]. This phenomenon has been previously observed in MSCs[59]. MSC exposure to IFN-γ has also been shown to enhance the expression of the enzyme indoleamine-2,3-dioxygenase[48,61], a critical regulator of MSC-mediated suppression of T-cell proliferation in vitro that depletes tryptophan pools[62,63]. Alterations of tryptophan catabolism through the kynurenine pathway were indeed detected by TENGi MS, with a FC for kynurenine (feature #6, m/z 209.0927) of 15.6 (Table 1 and Fig. 6b). At the same time, a significant decrease in tryptophan (Table 1, feature #8, FC = 0.0066, Fig. 6b) was also detected. Vunjak-Novakovic et al. observed an increase in kynurenine in IFN-γ-stimulated MSCs, but failed to detect the decrease of tryptophan[59]. The 5-HTP and 3-MDI pathways also showed significant alteration upon IFN-γ stimulation, which had not been previously reported as associated with MSC's inflammatory responses. 5-HTP is the immediate precursor of serotonin, which has been shown to inhibit apoptosis in multiple immune cell types under inflammatory conditions[64]. In terms of 3-MDI, reports exist in the literature regarding its detection in rat urine[65], serum[66] and human breath[67], but its function in MSCs upon inflammatory stimulation is yet unclear.

With only 80,000 cells and <1 min detection time, TENGi sub-nanoliter metabolomics showed promising performance for small cell number metabolomics in proof-of-principle studies. TENGi could likely benefit metabolomics studies of other rare cell types without the need for many rounds of culture. It not only can be applied to cases where relaxing sample volume requirements can facilitate both bioresearch and biomaterial manufacturing, as with MSCs, but is also highly suitable for cases where collecting large amount of sample is difficult, as with EBC.

**Polarity switching experiments**. As TENG generates both positive and negative pulses alternatively, coupling of this technique with the rapid polarity switching capability of modern mass spectrometers can realize the detection of both positive and negative MS information in one experiment. This experiment allows for coverage of both basic and acidic metabolites without additional experiments and sample consumption, further increasing the information density obtained per experiment or per sample load. A proof-of-concept experiment with liver lipid extract is shown in Fig. 9. Both negative and positive spectra were collected in the same run with only sub-microliter sample loading and sub-nanoliter sample consumption.

## Discussion

Unlike hypothesis-driven studies targeting a limited set of analytes, non-targeted metabolomics strives to monitor hundreds to thousands of molecular species in a single experiment. Non-targeted metabolomics can be regarded as a hypothesis-generating process that holds great promise for discovering previously unknown biomarkers and biochemical pathway alterations.

Classical metabolomics tools, such as LC-MS and nuclear magnetic resonance spectroscopy, require relatively large sample volumes, making many sample types not suitable. TENGi MS, a pulsed technique, makes full use of each spray event by consuming only sub-nanoliters of sample. The stock-sample volume needed in TENGi MS metabolomics is at least 10–20 times lower than that needed for conventional GC/LC-MS metabolomics. The lower sample volumes in TENGi MS make it an attractive option for rare sample metabolomics studies in either targeted or non-targeted modes.

Synchronization of the spray pulse with mass spectrometer scans and coupling TENGi with rapid polarity switching can be used to increase sample utilization and maximize the information obtained from limited amounts of sample. TENGi's high sensitivity stems from the high-voltage output of the TENG itself, enabling the detection of metabolites with significant S/N ratio

**Table 1 Annotation of the 14 most significant features used in discriminating stimulated vs. unstimulated MSCs.**

| # | $m/z$ | Fold change | $p$-Value | Identity | Formula | Adduct type[a] | Theoretical $m/z$ | Error (p.p.m.) | Chemical annotation method |
|---|---|---|---|---|---|---|---|---|---|
| 1 | 123.0552 | 3.2E + 00 | 4.4E − 03 | Niacinamide | $C_6H_6N_2O$ | $[M + H]^+$ | 123.0553 | 0 | MS/MS matched with database |
| 2 | 137.0465 | 1.5E − 01 | 2.1E − 02 | Hypoxanthine | $C_5H_4N_4O$ | $[M + H]^+$ | 137.0458 | 4 | MS/MS matched with database |
| 3 | 146.0598 | 2.2E + 01 | 1.8E − 05 | 3-Methyldioxyindole | $C_9H_9NO_2$ | $[M + H-H_2O]^+$ | 146.0600 | 1 | IM-MS/MS main fragments matched with in silico fragments.[b] |
| 4 | 163.9777 | 2.4E + 00 | 5.4E − 03 | Taurine | $C_2H_7NO_3S$ | $[M + K]^+$ | 163.9778 | 1 | Potassium adduct of taurine chemical standard could not be fragmented; accurate mass matched with standard |
| 5 | 170.0321 | 2.3E + 00 | 2.0E − 02 | Creatine | $C_4H_9N_3O_2$ | $[M + K]^+$ | 170.0326 | 3 | MS/MS matched with database |
| 6 | 209.0927 | 1.6E + 01 | 7.6E − 04 | L-Kynurenine | $C_{10}H_{12}N_2O_3$ | $[M + H]^+$ | 209.0921 | 3 | MS/MS matched with chemical standard MS/MS |
| 7 | 220.1077 | 3.9E + 02 | 2.5E − 15 | 5-Hydroxy-l-tryptophan | $C_{11}H_{12}N_2O_3$ | $[M + NH_4-H_2O]^+$ | 220.1080 | 1 | MS/MS manually matched |
| 8 | 227.0793 | 6.6E − 03 | 4.6E − 14 | L-Tryptophan | $C_{11}H_{12}N_2O_2$ | $[M + Na]^+$ | 227.0791 | 1 | MS/MS matched with chemical standard MS/MS |
| 9 | 232.0762 | 2.6E + 01 | 1.0E − 05 | Cysteinyl-Glutamine | $C_8H_{15}N_3O_4S$ | $[M + H-H_2O]^+$ | 232.0756 | 2 | N/A |
| 10 | 243.0745 | 2.1E + 02 | 2.9E − 10 | 5-Hydroxy-l-tryptophan | $C_{11}H_{12}N_2O_3$ | $[M + Na]^+$ | 243.0740 | 2 | MS/MS matched with chemical standard MS/MS |
| 11 | 247.0476 | 2.6E + 02 | 1.5E − 09 | L-Kynurenine | $C_{10}H_{12}N_2O_3$ | $[M + K]^+$ | 247.0479 | 1 | MS/MS partially matched with L-Kynurenine standard MS/MS |
| 12 | 265.1559 | 2.2E + 00 | 1.8E − 02 | 5-Tetradecenoic acid (or other isomers) | $C_{14}H_{26}O_2$ | $[M + K]^+$ | 265.1564 | 2 | N/A |
| 13 | 377.3017 | 4.2E + 00 | 2.1E − 02 | 2/3-hydroxydodecanoyl carnitine | $C_{19}H_{37}NO_5$ | $[M + NH_4]^+$ | 377.3010 | 2 | N/A |
| 14 | 785.5781 | 1.8E − 01 | 3.7E − 03 | PC (35:4) | $C_{43}H_{78}NO_8P$ | $[M + NH_4]^+$ | 785.5803 | 2 | MS/MS matched with database |

FC values were calculated as the stimulated/unstimulated IS-normalized abundance ratio. The two-tailed Student's t-test was used to calculate p-values. These were calculated on the final dataset obtained after G-log transformation and autoscaling through MetaboAnalyst. Source data are provided as Source Data file.
[a]In positive-ion mode nanoESI, $[M + H]^+$, $[M + Na]^+$, and $[M + K]^+$ are very common adduct types, as also shown in Fig. 2d–f and Supplementary Fig. 6. The $[M + NH_4]^+$ adducts detected here mainly arise due to the addition of $NH_4Ac$ during MSC processing, which significantly promotes $[M + NH_4]^+$ adduct formation (see also results in Supplementary Fig. 16).
[b]For feature #3, ion mobility (IM) MS/MS was used because precursor ion co-selection happened in DI MS/MS experiments.

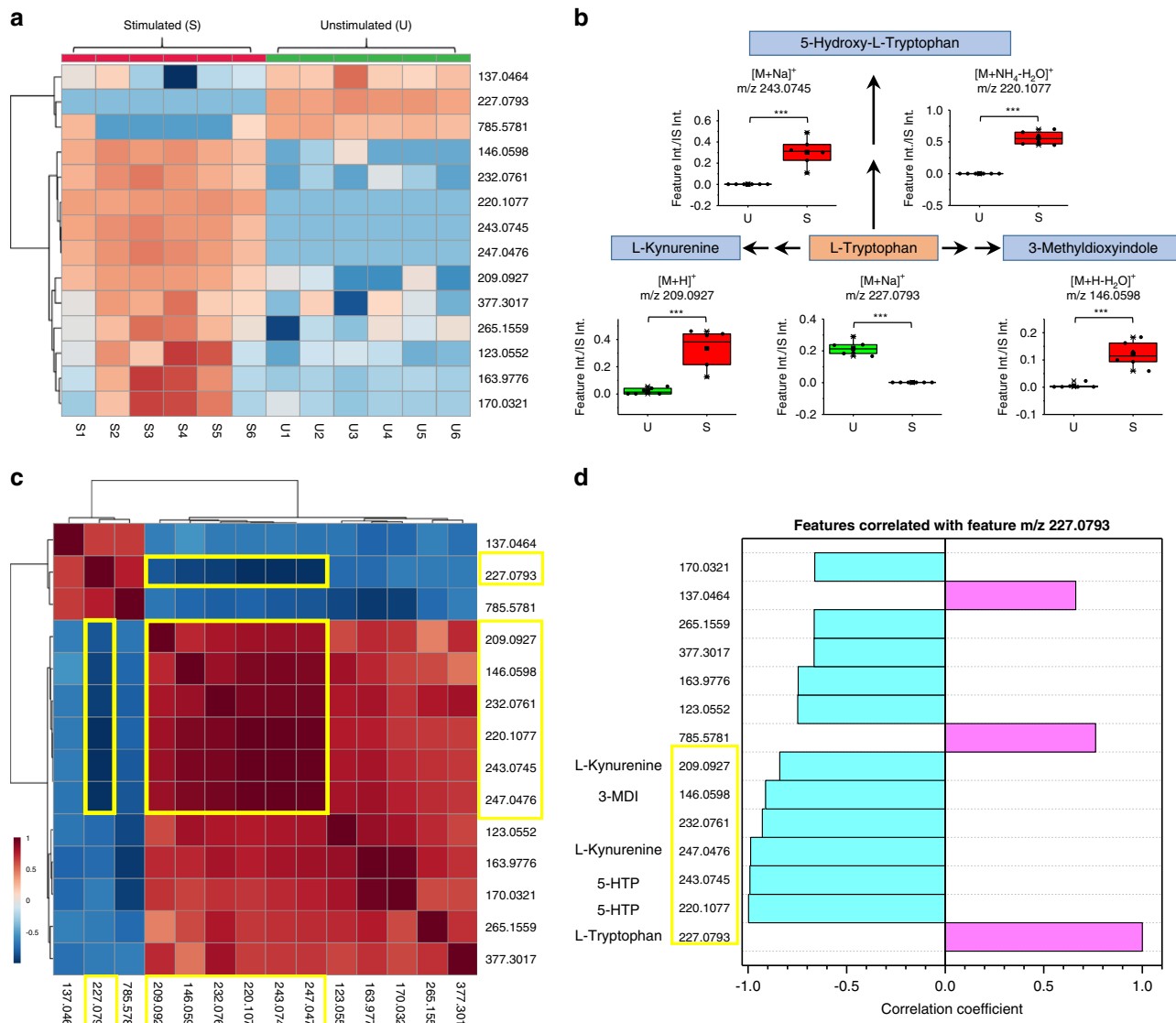

**Fig. 8 IFN-γ stimulation affects tryptophan metabolism in MSCs through three catabolic pathways. a** Heat map of the top 14 most important features. **b** Pathway analysis results, showing three catabolic pathways are affected. Box plots shows the alterations in these tryptophan-related metabolites. Differences between the two groups were evaluated using the two-tailed Student's *t*-test (***p-value < 0.001, detailed p-values of each feature: top left to right: 2.9E − 10, 2.5E − 15; bottom left to right: 7.6E − 04, 4.6E − 14, and 1.8E − 05. These are also provided in Table 1. *P*-values were calculated on the final dataset after G-log transformation and autoscaling through MetaboAnalyst). Source data are provided as Source Data files. The lower, middle, and upper lines in the box plots (in **b**) correspond to 25th, 50th, and 75th percentile. The whiskers extend to the most extreme data point within 1.5 IQR. *n* = 6 biologically independent samples were included in each group. **c** Pearson's correlation map for the 14 feature set. **d** Pearson's correlation coefficient between the 14 features of interest relative to *m/z* 227.0793 (tryptophan). Plots **c** and **d** show that there are six features closely correlated with *m/z* 227.0793 (tryptophan), five of which are from tryptophan catabolic products shown in **b** pathway analysis. Yellow squares indicate the metabolites that are highly correlated. Note 5-HTP has two ion forms as does L-kynurenine. Due to space limitations, **b** only shows the box plot of [M + H]+ (*m/z* 209.0927), the [M + K]+ (*m/z* 247.0476) ion is shown in Supplementary Fig. 11.

gains over conventional nanoESI techniques. Tiny volume TENGi metabolomics pilot studies of both EBC and cultured MSC samples illustrate the possibilities of the approach when coupled with high-resolution accurate mass spectrometers. Other limited-volume sample types, such as exosomes, tissue-derived cells, needle biopsies, tears, and sweat, could also benefit from the sub-nanoliter metabolomics strategy presented here. Relaxing sample volume requirements will also facilitate collection of more granular time points in animal studies without causing unnecessary harm.

Overall, TENGi MS metabolomics has shown promise for the analysis of very small sample volumes. A notable shortcoming of

the current configuration, however, is that the sample was loaded manually, but this issue could be readily solved in the future by using robotic automation. It is also important to note that TENGi MS is a DI metabolomics approach, and similar to other DI method, it lacks isobaric resolution. Coupling TENGi to IM-MS[68] would be a promising way to further increasing coverage, and be applied to even broader range of rare sample metabolomics studies.

## Methods

**TENGi nanoESI setup**. All nanoESI emitters were purchased from New Objective (Econo 12-N, 1 ± 0.5 μm) and had a length of 5.5 ± 0.1 cm, OD: 1.2 mm, ID 0.69

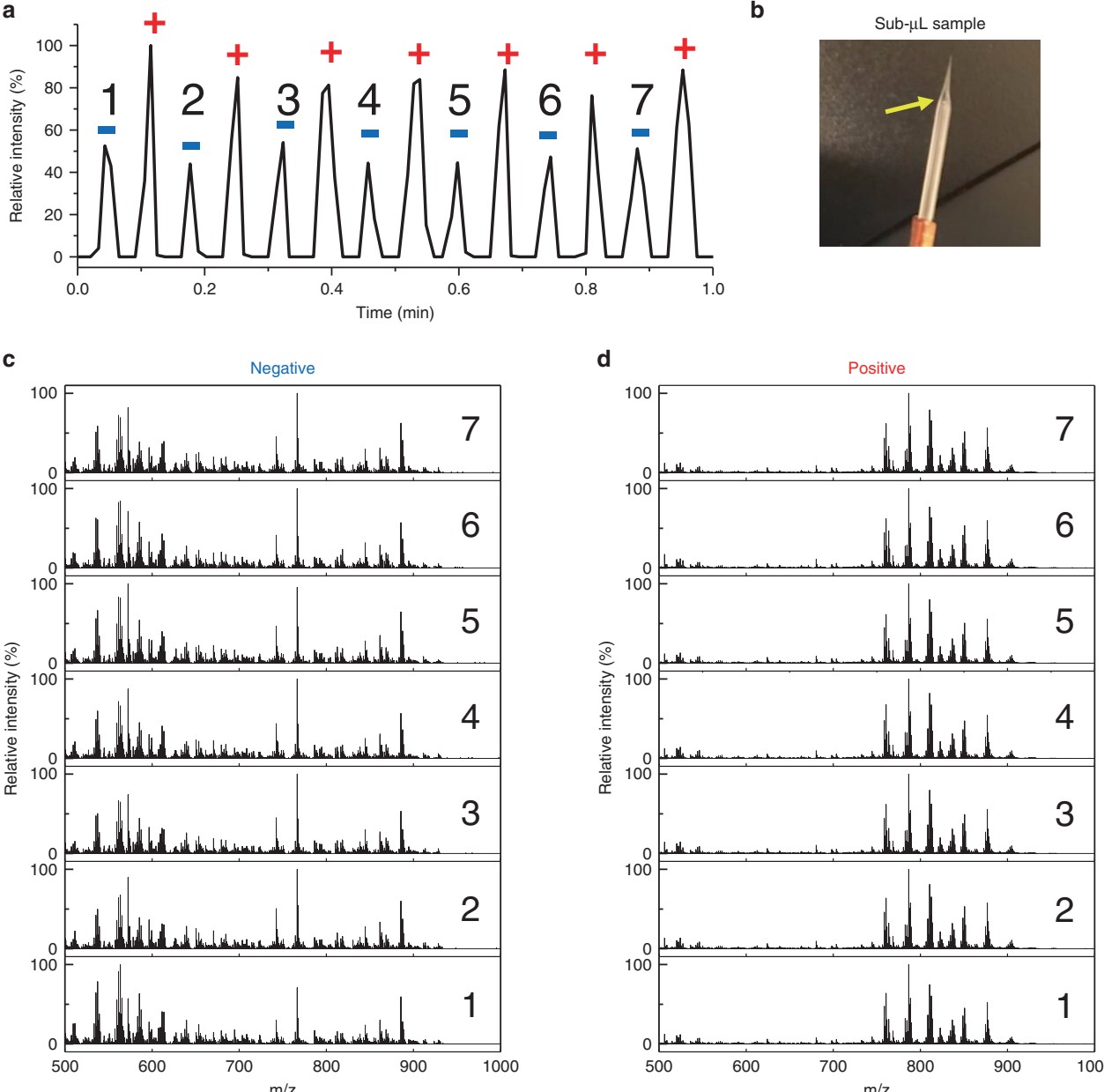

**Fig. 9 Polarity switching TENGi MS to obtain both positive and negative signals in the same experiment with sub-nanoliter sample consumption.** The sample tested was a 50 μg mL$^{-1}$ liver lipid extract. **a** TIC of TENGi coupled with polarity switching, Negative (blue "−" sign) and positive (red "+" sign) signals are detected alternatively. **b** Detail of emitter with 0.8 μL sample loaded. Yellow arrows indicate the position of the sample solution in the emitter. **c** Mass spectra for negative pulses; **d** mass spectra for positive pulses.

mm, emitter size of 1 ± 0.5 μm. A custom cartridge was fabricated to serve as a nanoESI emitter holder to ensure optimal positioning with respect to the mass spectrometer inlet (Supplementary Fig. 2). The semi-enclosed cartridge ensured that ionization occurred without disruption from surrounding air currents. The copper electrode used to induce electrospray was wrapped on the outside of the nanoESI emitter and connected to one of the TENG outputs. The other TENG output was grounded. A home-made TENG mechanical actuator was used for EBC metabolomics experiments; a linear motor was used for the MSC experiments. Typical TENG operation frequencies were in the 0.5–1 Hz range.

**Mass spectrometry.** Two mass spectrometers were used—a Waters Synapt G2s quadrupole IM time-of-flight and a Thermo Orbitrap Q-Exactive. All TENGi MS characterizations and EBC metabolomics experiments were carried out on the Synapt G2s in negative-ion mode. Synapt ion source parameters were as follows: capillary temperature of 120 °C, sampling cone voltage at 40 V, and source offset at 80 V. MS/MS feature annotation used a low-mass resolution value of 10, and collision energy (CE) of 10–40. MSC metabolomics experiments were carried out on the Orbitrap Q-Exactive, with the following ion source parameters: capillary

temperature of 300 °C, S-lens at RF level 50, resolution of 17,500, max injection time of 200 ms, and automatic gain control target of 5e6. We chose not to operate at a higher Orbitrap mass resolution setting and maximize the Orbitrap cycle time so as to obtain sufficient points across TENGi peaks. Orbitrap MS/MS experiments used an $m/z$ window of 1, and a normalized CE value of 10–30.

**TENGi MS experiments.** For each set of experiments, 0.8 μL of sample solution was loaded into the nanoESI emitter with an Eppendorf GELoader. The end of the emitter was immediately sealed with Parafilm to prevent evaporation. This was the minimum volume that could be loaded using a pipette, but smaller samples could be utilized using alternative sample loading approaches. Twenty (for MSC experiments) to forty (for EBC experiments) TENGi pulses were obtained for each sample. With the MS tune settings utilized, six to seven scans were acquired per pulse with a pulse width variation of 8.9% (Supplementary Fig. 9b, c). Outlier pulses were discarded based on the extracted ion trace pulse areas for the IS, and three (for MSC experiments) or five (for EBC experiments) of the remaining pulses were randomly selected to generate the representative spectrum for the corresponding sample. Outlier rejection was based on a ±20% interval centered around

the median area of the IS EIC. More details are provided in Supplementary Fig. 21. Collecting about 20–40 pulses were to get enough statistics to help identify and avoid extreme pulses; only use any 3–5 of them was to show that in cases where sample is extremely rare, only several pulses of signal is enough to do metabolomics. These data processing steps were carried out using Masslynx (Waters) or Xcalibur (Thermo Fisher). Negative-ion mode in the 50–750 $m/z$ range was used for EBC experiments and positive-ion mode in the 67–1000 $m/z$ range for experiments with MSCs.

**EBC sample preparation**. EBC from 11 CF patients (Supplementary Table 1) with abnormalities in glucose homeostasis (as defined by having CF IGT on an oral glucose tolerance test) was collected both fasting and 2 h following ingestion of a glucose drink. They were termed as Pre and Post, respectively. Of these 11 subjects, one did not have an adequate sample for the Pre and another did not have a sample for the Post study giving a total of 20 samples analyzed. EBC sample collection was carried out with informed consent from the donors and followed the guidelines approved by the Georgia Institute of Technology and the Emory University Institutional Review Boards (approval number IRB00000372). Samples were collected with an R-Tube collector (Respiratory Research, Inc., Austin, TX, USA). After collection, samples were immediately frozen at −80 °C until processed. A total of 20 EBC samples (10 for each group), each 50 μL in volume before concentration, were phenotyped. A pooled sample, which was used as a QC, was prepared by taking 5 μL from each EBC sample and mixing the aliquots together. Then, the 20 samples, together with the pooled QC sample and the blank sample (containing only ultrapure water) were lyophilized at −40 °C and 100 mTorr for 24 h in a VirTis Benchtop freeze-drier (LP Industries, Stone Ridge, NY, USA). Residues were then reconstituted in 9 μL of methanol/water 1:9 (v : v) with 1 × $10^{-6}$ M $^{13}$C tyrosine spiked in. This resulted in a fivefold concentration factor.

**MSC culture and priming**. Bone marrow-derived MSCs (RoosterBio, Inc., Lot #000139) were expanded for two passages in culture after being received. They were frozen in ~5 × $10^5$ aliquots in Cryostor CS10 freeze media (BioLife). Frozen aliquots were revived and plated in tissue culture polystyrene flasks (Corning) for 3–4 days prior to seeding onto test surfaces. MSCs were cultured in low-glucose Dulbecco's modified Eagle's medium (Gibco) supplemented with 10% fetal bovine serum (Atlanta Biologicals, lot E16063), and 1% antibiotic/antimycotic solution (Gibco). Once confluent, MSCs were washed with sterile-filtered phosphate-buffered saline (PBS, Thermo Fisher) and detached from flasks using TrypLE express (Thermo Fisher). Dissociated cells were counted using a hemacytometer and replated at 13,000 cells/cm$^2$ in T-75 tissue culture flasks. After overnight incubation, MSCs then were exposed to 48 hours of culture media (control conditions), or culture media supplemented with 50 ng/mL IFN-γ (Thermo Fisher). MSCs were then washed with PBS and trypsinized from flasks using TrypLE express (Thermo Fisher), and the number of collected cells counted using a hemocytometer. MSCs were then resuspended in 155 mM ammonium acetate (Fluka) at a concentration of $1.6 × 10^6$ cells/mL and aliquoted into 50 μL samples ($8 × 10^4$ cells per aliquot). Cells were then quenched by adding 200 μL MeOH into each sample vial and stored at −80 °C until metabolite extraction.

**MSC metabolite extraction**. Frozen cells were subject to three freeze-thaw cycles, with liquid nitrogen for freezing and ice-water sonication for thawing. Cell samples were then centrifuged at 21.1 × $g$ for 5 min to precipitate proteins. From the supernatant, 200 μL was transferred into a new vial for lyophilization. The pooled QC sample was formed by mixing 30 μL of each sample. All cell extracts and the QC sample were then lyophilized at −40 °C and 100 mTorr for 24 h in a VirTis Benchtop free-drier (LP Industries, Stone Ridge, NY, USA). Residues were reconstituted in a $5.9 × 10^{-5}$ M $^{13}$C-phenylalanine methanolic solution to a final volume of 10 μL (for samples), and 18 μL (for QCs). The run sequence used after samples were prepared is shown in Supplementary Fig. 14. MS experiments were carried out in positive-ion mode, with 0.8 μL of sample for each test. Stimulated and unstimulated MSC samples were subject to TENGi MS in an alternating, randomized order. Triplicate pooled QCs were run every six MSC samples to monitor the technical quality of the results.

**Data processing and statistical analysis**. The main data processing steps were inspired by those reported by Viant et al.[7,24] and are depicted in Fig. 2c. The peak list from each TENGi mass spectrum was first exported to Excel. Mass peaks with abundances lower than 2E3 (Synapt G2s) or 1.5E4 (Orbitrap QE) were removed from the dataset. Following alignment, all peaks were normalized against the IS. Then, a series of filters were applied: (1) blank filtering was applied to remove signals from biological samples that had normalized abundances lower than three times that of the blank sample; (2) prevalence filtering by only retaining peaks present in at least 80% of the biological samples, (3) filtering peaks with 80% missing values or not detected in the QC sample. The dataset then was subject to missing value imputation using half of the minimum positive value in the original data as the replacement. Further spectral cleaning was performed to remove peaks which had RSD$^{QC}$ values >30%. During data analysis, one sample from the Post group was deemed an outlier as it had more than 20% missing values (Supplementary Fig. S5)[24] and was removed from the dataset. The dataset was then

subjected to MetaboAnalyst for G-log transformation, autoscaling, and multivariate analysis. PCA, PLS-DA, and two-tailed Student's $t$-test with unequal variance were performed in MetaboAnalyst (https://www.metaboanalyst.ca/MetaboAnalyst/home.xhtml, a widely used on-line platform for metabolomics analysis[69]). oPLS-DA modeling was carried out with the MATLAB PLS Toolbox (v. 8.0), also a standard tool for multivariate analysis. Leave-one-out cross-validation was used for classification. After strict data filtering, ~158 features were retained for EBC samples, but only those that had putative IDs in Lipid Maps (https://www.lipidmaps.org/), Metlin (https://metlin.scripps.edu/), and the Human Metabolome DataBase (http://www.hmdb.ca/), with an error of 10 p.p.m. were preserved for multivariate analysis (72 features). For TENGi MSC metabolomics, 389 features passed the strict data filters and used for multivariate analysis. Among them, 135 were tentatively identified based on their accurate masses (<5 p.p.m.). These results are included in the Source Data file. Pearson correlation analysis shown in Fig. 8c, d were carried out with MetaboAnalyst using the g-log transformed and auto-scaled dataset.

**Statistics and reproducibility**. The TENGi method characterization experiments were all repeated at least three times. The two metabolomics studies were only carried out once because of the difficulty in collecting samples, but three technical replicates (injections) were conducted for all the tested samples. All attempts at replications were successful.

**Reporting summary**. Further information on research design is available in the Nature Research Reporting Summary linked to this article.

## Data availability

Data generated in this work are available through the NIH Metabolomics Workbench (http://www.metabolomicsworkbench.org/) under project ID PR000988 with study ID ST001437 for EBC data and study ID ST001438 for MSC data. All other data are available from the corresponding author on reasonable request. Source data are provided with this paper.

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

## Acknowledgements
F.M.F. acknowledges support from 1R01CA218664-01 and NIH 1U2CES030167-01. F.M.F. and J.S.T. acknowledge support from the CMaT NSF Research Center (EEC-1648035). A.A.S. acknowledges support from FDA R01FD003527-01. Y.L. acknowledges Dr. Wenbo Ding and Dr. Zhiyi Wu from Professor Wang's group for their assistance in TENG device characterization.

## Author contributions
Y.L. and F.M.F. designed the study. Y.L. performed most of the experiments and data analysis. M.B. contributed to the analytical characterization of TENGi MS and mass spectrometer parameter optimization. C.W., H.G., and Z.L.W. fabricated the TENG device and performed the mechanical and electrical characterization, as well as theoretical simulations of the TENG device. D.H. contributed to metabolomics data processing. G.D. and J.S.T. prepared MSCs. A.A.S. provided EBCs of CF patients. Y.L. and F.M.F. wrote the manuscript. All authors discussed the results and commented on the manuscript.

## Competing interests
The authors declare no competing interests.
