## [Peer Review File · Nature Communications]

REVIEWER COMMENTS

Reviewer #1 (Remarks to the Author):

Overall, this manuscript presents an interesting technique addressing a specific concern within the community of micro-sampling for MS-based analysis. The authors attempt to straddle the technical report and application aspects but encounters the expected difficulty of being somewhat less satisfying than a fully novel technical report or a clearcut biological report. The scope of the work is ambitious and several of the concepts proposed by the authors are appealing exercises in metabolomics. However, the execution of the work is problematic on a number of levels, including questionable practices in statistics as well as the almost complete lack of validation of the very large number of predictions and claims made in the paper.

On the technical development side: On a general level I find the introduction and presentation of the method repetitive and somewhat superficial. It is hard to assess the systematic performance of the technique and actual benefit with respect to standard methodologies. More specifically:

- How do the authors accurately account for 100% ion utilization? This sentiment is repeated in the discussion, line 383-387, without any proof that TENGi uses all ions .
- How do the authors overcome in-source fragmentation to confirm that the metabolites ID'd weren't from high voltage conditions? and How is the voltage evenly applied to the spray solution?
- Is the sample size large enough for an accurate metabolome coverage? The authors should comment on this aspect. For example what is the coverage of central metabolism . It appears that there are too few metabolites in the PLS-DA dataset. Shouldn't be more representative metabolites that pass these standard volcano plot criteria. metabolites represented in Figure 4, the 15 might give the impression that this method has poor metabolite coverage and has to be well tuned to only produce a few ions.
- Ideally one would need to see more internal standards used to understand how a variety of functional groups perform under TENGi conditions.
- The adduct formation used for metabolite table T1 is interesting, it suggests that TENGi produces adducts formation more so than traditional ESI/nanoESI . The author should comment on this aspect.
- Line 373-376 would lead the reader to expect that TENGi is untargeted, but with the raw data shown in the manuscript that isn't clear. TENGi is still as biased as ESI, however, it would be desirable to better understand to what degree the ions that are generated are more specific to the technique.
- Typo in line 209 should indicate 3b and 3d (the correction).

- The authors should also avoid overstatements and be more precise including a comparison not only with cherry picked technologies that are clearly outperformed by theirs. Like in lines 263-274. There are several techniques which can work with way fewer cells and 1 million.

On the biological insights side:

The work contains a large number of predictions of potential metabolite characteristics of a disease. However, beyond some retrospective analysis of literature around a handful of cherry-picked individual predictions, there is no real indication of the accuracy of any of these predictions, rendering the predictions of little real value it seems.

Reviewer #2 (Remarks to the Author):

General comments:

The research around “tiny metabolomics” is timely and important. The approach presented, more specifically the ionization method allowing sub-nanoliter sample consumption is novel and deserves publication. While 0.8 μL sample volume are filled in the emitter, and only pL are consumed for analysis using the TENGi nanoESI approach, a traditional DI nanoESI MS would require 20-30 μL sample volume per well, out of which 10-20 μL are consumed upon analysis. TENGi nanoESI requires changing emitters per sample. Thus, the huge advantage of the traditional “nano” method is sample throughput and automation. Despite this, I think that the new development will have its place as there is an increased interest in multi-omics studies having to deal with very precious limited sample volumes. The emerging topic of single cell MS based-omics can profit from the technology as well.

The authors show adequate proof-of-principle applications.

From the perspective of an analytical chemists, a more thorough comparison with DI-nanoESI MS and/or conventional metabolomics would have been desirable. The authors claim, that compared to traditional DI nanoESI MS, TENGi nanoESI has better signal-to-noise (S/N) ratios, but then this is shown only for one compound. It would have been desirable to proof with complementary conventional approaches that discoveries are not severely limited due to the sub-pL sample consumption or on the contrary even improved (as the authors claim). This would have been possible by simple experiments with standards spiked into biological matrices. In my opinion the authors fail to proof this specific point.

Overall, I recommend publication following addition of experiments, minor changes and clarifications.

Specific comments:

Line 40 “volume-limited biospecimens include cerebrospinal fluid”

The sample volume of CSF samples is in the mL range! Thus I do not regard this as a prime example of “tiny metabolomics”.

The following statement (line 78-79) is not true: “However, conventional GC- or LC-MS-based cell metabolomics typically requires millions of cells.” Limits of detection of e.g. orbitrap technology using the conventional LC set up (2 mm column, flow rate 250-300 $\mu\text{L}/\text{min}$; injection volumes of μL) are in the pM to sub-nM range, which in turn enables to measure cell numbers of 10.000-100.000 cells on a routine basis.

The same holds true for the statement (line 280) “The analysis used only 80,000 cells per sample, which is more than an order of magnitude lower than that required for conventional LC-MS methods”

Such a cell number could be analysed by conventional approaches. In fact, the concentration based LOD/LOQ are the same or even better in conventional LCMS (LOD of this study zeptomol/pL corresponding to nM)

Figure 1: Could you please specify how many scans are recorded per pulse? What is the RSD of the lactic acid signal obtained from multiple pulses? Is there an estimate of the lactic acid concentration range? What is the RSD of the pulse width?

In the introduction, the authors state that the combination of pulsed MS measurements and pulsed ionization allows a 100% sample utilization, which seems indeed an intriguing idea. Did I understand this correctly that a “real synchronization” is a future aim and for now the cycle times are independently controlled, but adjusted to fit? The authors do not further comment on how many measurement cycles are performed per pulse. This information is hidden in the method part.

Table 1: please give significant numbers for fold change!!!

Line 364...”With only 80,000 cells and <1 min detection time,

I think this statement that the novel approach is superior compared to conventional metabolomics, should be rephrased, since it is not true in such a generalized manner. The new technology is absolutely superior in terms of sample consumption, but the study does not allow to draw this final conclusion of higher sensitivity and thus higher discovery potential. This would have required the validation of the method by conventional metabolomics. The authors come to this conclusion, simply, because key findings regarding in the MSC study have not been published before. This is remarkable but not a stringent argument with regard to detection power. There might be multiple reasons why previous studies failed to discover certain metabolic rearrangements.

All measurements have been carried out in negative mode. Is this a technological restriction of the ionization? Would pos/neg switching be possible? Please comment!

Line 422-424 “Twenty (for MSC experiments) to forty (for EBC experiments) TENGi pulses were obtained for each sample. Three (for MSC experiments) or five (for EBC experiments) of such pulses were selected based on the extracted ion trace pulse areas for the IS (1 μM 13 C-tyrosine), discarding any outliers to ensure maximum reproducibility”

This is my major concern, where clarification is needed! Have all data (the nice RSD values shown for QC samples and internal standards) been based on such a selection, 3-4 pulses out of 20/40? This is a very problematic data evaluation, where the statistical foundation of this outlier selection (the majority of pulses are outliers?) becomes very fragile. Please comment! This type of data selection makes the results rather arbitrary!

What is the RSD of the 40 pulses?

Reviewer #3 (Remarks to the Author):

The authors presented a new mass spectrometry the triboelectric nanogenerator inductive

nanoelectrospray ionization with making full use of each spray event by consuming only sub-microliters of sample. Different with the previous work, this manuscript focused on converging this device with exhaled breath condensates and showed the advancement in easy sample collecting. This manuscript is acceptable to Nature Communications after some modification. The detailed comments are as follows:

1. Even though the detailed part related to TENG is dealt in previous work (ref. 7), TENG part should be added to the manuscript. (e.g. fabrication process, simple principle, and basic electrical output)
2. The order of the figures is confusing due to the not matching with the order of the result part. The readability would be increased with a suitable relocation.
3. In Figure 1c and S3, the meaning of values on graph and more detailed explanation are required for easy understanding to readers.
4. In Figure S4, the authors showed the preserved samples with 10 days. However, how can we know the preserved samples show the same characteristic compared with 0 day sample?
5. In the part of Figure S6b, the elucidation of how to obtain the sensitivity and specificity needs to be supplemented.
6. There is no Figure 4b. It needs to be added to the manuscript in appropriate part.

7. In the starting part of the sentence dealing with Table S2, there is a typographic error and this punctuation needs to be modified.

We would like to thank the reviewers for their very insightful comments. We appreciate the time and dedication that is required to produce high quality peer reviews. We have taken the comments very seriously and have added a significant amount of new experiments, data, and figures to address the feedback you have given us, as summarized below.

Summary of changes:

1) Newly added figures and illustrations (marked in red in the revised manuscript):

Location	Newly added figures and illustrations	Contents
Main text	Figure 1b:	Schematic illustration of the SF TENG device used in this study
	Figure 1e:	Illustration of the charge generation mechanism in TENGi
	Figure 1g:	Illustration of sample consumption and utilization in conventional DC ESI/nanoESI and TENGi.
	Figure 2d-f	Behavior of four typical metabolite standards (Adenine, Lysine, Glucose, and LysoPC) ionized by TENGi and conventional DC nanoESI.
	Figure 3	Dilute liver lipid extract (25 µg/mL) mass spectra produced by both conventional DC nanoESI and TENGi.
	Figure 9	Polarity switching TENGi experiments to obtain both positive and negative signals in the same experiment with only subnanoliter consumption of a 0.8 µL load of sample.
Supporting Information	Figure S2	TENG charging mechanism illustration
	Figure S4e,f	TENGi mass spectra (blank subtracted) in both positive and negative ion modes after storage for 10 days.
	Figure S6	Comparison of TENGi and conventional DC nanoESI using four different types of common metabolites.
	Figure S7	In-source fragmentation experiments.
	Figure S8	Lactic acid calibration curve and quantification.

	Figure S9	TENGi pulse characterization: TIC, pulse width variation, lactic acid signal variation.
	Figure 15 b,c	TENGi MSC metabolomics PLS-DA loadings plot and top 50 PLS-DA VIP score plot.
	Figure S16	$[M+NH_4]^+$ formation in TENGi
	Figure S21	Outlier identification using internal standard.

2) Original figures and illustrations that have been renumbered or re-ordered (**marked in blue in the revised manuscript**):

	Old name	New name
Main text	Figure 1a right	Figure 1d
	Figure 1b	Figure 1c
	Figure 1c	Figure 1f
	Figure S1a b c	Figure 2 a b c
	Figure 1 d-f	Figure 2 g-i
	Figure 2	Figure 4
	Figure 3	Figure 5
	Figure 4	Figure 6
	Figure 5	Figure 7
	Figure 6	Figure 8
Supporting Information	Figure S1 d-f	Figure S5 a-c
	Figure S2	Figure S1
	Figure S5	Figure S10
	Figure S6	Figure S11
	Figure S7	Figure S12
	Figure S8	Figure S13
	Figure S9	Figure S14
	Figure S10 a,b	Figure S15 a, d
	Figure S11	Figure S17
	Figure S12	Figure S18
	Figure S13	Figure S19
	Figure S14	Figure S20

POINT-BY-POINT RESPONSE TO REVIEWER COMMENTS

Reviewer #1 (Remarks to the Author):

1. Overall, this manuscript presents an interesting technique addressing a specific concern within the community of micro-sampling for MS-based analysis. The authors attempt to straddle the technical report and application aspects but encounters the expected difficulty of being somewhat less satisfying than a fully novel technical report or a clearcut biological report. The scope of the work is ambitious and several of the concepts proposed by the authors are appealing exercises in metabolomics. However, the execution of the work is problematic on a number of levels, including questionable practices in statistics as well as the almost complete lack of validation of the very large number of predictions and claims made in the paper.

On the technical development side: On a general level I find the introduction and presentation of the method repetitive and somewhat superficial. It is hard to assess the systematic performance of the technique and actual benefit with respect to standard methodologies.

We apologize for the insufficient explanation provided in the original manuscript. A number of additional experiments has been conducted and data have been added to the revised manuscript so as to address the issues brought up by the reviewers. The newly added contents include further explanation of TENG and TENGi's technical characteristics, TENGi's advantages over conventional DC nanoESI using 4 metabolite standards, metabolomics studies using liver lipid extract, *etc.* Major changes are summarized in the tables above, and are discussed point-by-point below when addressing each of the specific comments.

To address the reviewer's point that "On a general level I find the introduction and presentation of the method repetitive and somewhat superficial", a number of additional experiments have been conducted, and data have been added to better demonstrate the TENGi technique. The newly added contents include further elucidation of TENG and TENGi's technical characteristics, TENGi's advantages over conventional DC nanoESI in sample analysis using 4 typical metabolite standards, metabolomics studies using liver lipid extract, *etc.* Detailed changes are summarized in the tables above, and will be discussed one by one below when addressing each of the comments.

As regard to the statistical treatment of the data, we have used accepted practices that are well documented in the literature, both for data processing and data analysis. We apologize that the approaches were not well described. In terms of data processing, we implemented strict filtering to limit the number of features in the final dataset and avoid over interpretation of the biological findings. Similar data processing procedures have also been used by Viant *et al.* in *Scientific data*, 1, 140012 (2014) and *Nature Protocols* 12, 310-328 (2016)). In terms of the multivariate analysis of the data, we chose PLS-DA and volcano plot for the selection of the most significant features as these are the choices of the metabolomics community in numerous publications. The analysis was carried out by MetaboAnalyst (<https://www.metaboanalyst.ca/MetaboAnalyst/home.xhtml>), a widely used on-line platform for metabolomics data analysis (*Nature protocols*, 6, 743-760 (2011)), and by the MatLab PLS-DA Toolbox, which is also a standard tool for multivariate analysis. We now have added this information to the Methods section.

Both of the metabolomics studies presented in the manuscript are seen as proof-of-concept studies geared at showing the capabilities of TENGi for "tiny metabolomics", but the intention was not

to derive very strong biological conclusions, as larger cohorts and more biological assays would be needed for that. To emphasize the nature of this work, we have now added statements at the end of each of the two metabolomics study sections clarifying this specific point.

More specifically:

2. - How do the authors accurately account for 100% ion utilization? This sentiment is repeated in the discussion, line 383-387, without any proof that TENGi uses all ions.

We apologize for the misleading statement. We have now reworded the Introduction so it states *“conventional ESI/nanoESI DI MS rarely achieves 100% duty cycle when coupled with pulsed mass analyzers. The result is unnecessary sample waste. Pulsed ion sources are more compatible with pulsed mass analyzers”*. These statements are to indicate that the TENGi’s pulsed characteristic better matches with the intermittent nature of Orbitrap/TOF mass analyzers. Therefore, higher duty cycle/sample utilization can be reached. To better illustrate this point, we have also added an illustration in the main text as Figure 1g (also shown below). Conventional DC ESI/nanoESI supplies sample continuously but only a small fraction is utilized by the mass analyzer. A large fraction of the sample is wasted. In the case of TENGi, each spray event can be better utilized to generate MS information, thus largely minimizing sample waste during analysis. This point has also been added to the Results section corresponding to Figure 1 as follows: *“An additional advantage is that TENGi’s pulsed characteristics are a better match to the intermittent nature of pulsed mass analyzer such as Orbitraps and time-of-flight. As is demonstrated in Figure 1g, conventional DC ESI/nanoESI supplies sample continuously, but only a fraction is utilized by the mass analyzer. A large fraction of the sample is wasted when the analyzer is busy scanning ions. In the case of TENGi, by synchronizing the ionization and mass analyzer cycles, each spray event can be more fully utilized to generate MS information, thus largely minimizing sample waste during analysis.”*

Figure 1g. Illustration of sample supply and utilization in conventional DC ESI/nanoESI and TENGi. Because the mass spectrometer intermittently traps and scans ions, TENGi can be synchronized with these events so as to achieve higher sample utilization.

3. – 1) How do the authors overcome in-source fragmentation to confirm that the metabolites ID’d weren’t from high voltage conditions? and 2) How is the voltage evenly applied to the spray solution?

1) In-source fragmentation is a confounding factor in metabolomics experiments as many metabolite can yield neutral losses that produce stable ionic species. In-source fragmentation typically occurs in the differentially-pumped regions of the atmospheric pressure interface of the mass spectrometer, where voltages are applied that exceed those needed for ion transfer without activation. A number of parameters affect the transport dynamics from the atmospheric pressure ion source to the vacuum mass analyzer, including those that affect supersonic jet expansion, the intermediate pressure region voltage settings, and the ions own internal energy (*Mass Spectrom. Rev.* 24, 566-587 (2005)). In this review, source voltage was not listed as a major factor for in-source fragmentation.

To assess whether TENGi causes extra in-source fragmentation compared to conventional DC nanoESI, we tested two common metabolites that have been reported to suffer from in-source fragmentation (guanosine and citruline, *Anal. Chem.* 87, 2273-2281 (2015)). Results showed that the in-source fragmentation extent (as measured by the in-source fragmentation ratio) using TENGi was slightly lower than for DC nanoESI, suggesting that although TENG provides much higher voltage than DC nanoESI, no extra in-source fragmentation is induced. This is likely because a) TENG's high voltage is transient and only a limited number of charges are generated per cycle (1.37 μC , Figure 2a) and b) the high voltage is applied indirectly to the sample solution. For comparison purposes, we also tested an approach named "0 V ionization" that consisted in loading about 1 μL of sample into a capillary directly contacting the MS inlet. In this scenario, ionization occurs through an aerodynamic breakup mechanism as discussed in *Anal. Chem.* 87, 6786-6793 (2015). These experiments showed that in-source fragmentation is more serious under "0V ionization" conditions than for both DC nanoESI and TENGi, indicating other factors may play bigger roles in in-source fragmentation than needle voltage does. These results have been now also been added to Figure S7, as shown below.

Figure S7. Investigation of the extent of in-source fragmentation. Three different ionization methods were compared: a) conventional DC nanoESI (1.5 kV); b) TENGi; 3) "0 V" inlet ionization. Two metabolites, guanosine and citruline, which have been reported to suffer from in source fragmentation were tested. The fragmentation ratios, calculated as percent of the precursor ion abundance, are shown in d) and e).

2) To address this point, we have added a new illustration (Figure 1e) to better describe our understanding of the charge generation and ionization mechanisms in TENGi. In this technique, as a type

of contactless ESI (*Anal. Chem.* 84, 7422-7430 (2012); *Angew. Chem. Int. Ed.* 50, 2503-2506 (2011); *Angew. Chem. Int. Ed.* 50, 9907-9910 (2011)), charges are generated mainly by electrostatic field-induced molecule polarization, ionization and charge separation. Prior to TENG actuation, the analytes in solution are evenly distributed to satisfy the charge balance. Once the TENG is actuated, electrostatic fields are created in C2 and C3 (see figure below), spray generated and ionization occurs. This discussion has now been added to the Results section, when describing Figure 1e.

Figure 1e. Charge generation mechanism in TENGi.

4. -1) Is the sample size large enough for an accurate metabolome coverage? The authors should comment on this aspect. 2) For example what is the coverage of central metabolism. It appears that there are too few metabolites in the PLS-DA dataset. Shouldn't be more representative metabolites that pass these standard volcano plot criteria. metabolites represented in Figure 4, the 15 might give the impression that this method has poor metabolite coverage and has to be well tuned to only produce a few ions.

1) TENGi's strength is for volume-limited sample characterization. In such cases, TENGi is an excellent option as it only consumes sub-nanoliters of sample and is highly sensitive. To better demonstrate that TENGi has the potential to achieve higher chemical coverage than conventional DC nanoESI, we performed extra experiments on a very dilute liver lipid extract ($25 \mu\text{g mL}^{-1}$). Results for these experiments are shown in the new Figure 3. In both positive and negative ion full MS analysis (Figure 3a and 3e), TENGi generated much higher ion abundances than conventional DC nanoESI. Most abundant species were detected by both DC nanoESI and TENGi. However, lower abundance species, such as the ion at m/z 800.616 (Figure 3b) and m/z 919.555 (Figure 3f), were not detectable in DC nanoESI. TENGi was still able to detect these species, generating sufficient ion current to also carry out MS/MS experiments (Figure 3c and 3g). In terms of feature numbers detected, TENGi produced 300 more features than DC nanoESI in positive ion mode (Figure 3d); in negative mode, the number of features detected by TENGi was almost twice that of DC nanoESI (Figure 3h). These findings have been added to the main text describing Figure 3.

Figure 3. Dilute liver lipid extract ($25 \mu\text{g mL}^{-1}$) analyzed by both conventional DC nanoESI and TENGi MS. a-d) Positive ion mode MS comparison: a) full MS, b) enlarged view of m/z 800.616 that was detected by TENGi but not by DC nanoESI, c) TENGi MS/MS identification of the species with m/z 800.616; d) Number of peaks detected in positive ion mode in the 700-1000 m/z range. e-h) Negative ion mode MS comparison: e) full MS, f) enlarged view of m/z 919.555 that was detected by TENGi but not by DC

nanoESI. g) MS/MS identification of the species with m/z 919.555; h) Number of peaks detected in negative ion mode in the 700-1000 m/z range.

2) The reviewer also poses a question regarding the number of metabolites observed in PLS-DA dataset. We assume the reviewer is referring to the feature numbers shown in the PLS-DA VIP score plot (current Figure 7c). In that plot, we chose to show only the top 20 most important ones to make the plot more readable. We have now added the corresponding PLS-DA loadings plot (which shows the distribution of all the 389 features) and the top 50 PLS-DA VIP score plot in Supporting Information as Figure 15b and 15c, respectively. We have also added a clarification to the legend of Figure 7c, to make this point more clear.

How many features pass through volcano plot depends on several factors, including the nature of the sample, the effect sizes between the two biological conditions compared, etc. We have 389 features in the final dataset. We tentatively annotated 135 of them based on their accurate mass (<5ppm). These results are included in Supporting Datasheet-1. Among them are a number of amino acids and their metabolites, purines and their metabolites, lipids etc. As we used positive mode for the MSC metabolomics study, most of the TCA cycle metabolites were not detected. Two glycolysis metabolites are found in this list, glucose and fructose 1,6-bisphosphate. These results show that coverage of the TENGi method is not limited. These discussions have also been added under Table S4.

It is also important to note that TENGi MS is a direct infusion metabolomics approach, and like other direct infusion method, it lacks isobaric resolution. Coupling TENGi to ion mobility-MS would be a promising way to further increasing coverage and be applied to even broader range of rare sample metabolomics studies. We added this statement in the Discussion section.

Figure S15. Comparison of interferon- γ (IFN- γ) “stimulated” and “unstimulated” MSC groups. b) PLS-DA corresponding loading plot, c) top 50 PLS-DA VIP features;

5. - Ideally one would need to see more internal standards to understand how a variety of functional groups perform under TENGi conditions.

We agree with this point, and we have now added additional experiments with more standard compounds spanning a variety of functional groups. We chose to use four typical metabolite standards to compare TENGi and conventional DC nanoESI: adenine, lysine, glucose and 18:1 LysoPC. The results have been added to the manuscript as Figure 2d-f and Figure S6, with corresponding discussion therein. In terms of adduct formation, both methods behave similarly, but the ion abundance in TENGi is higher (Figure 2d and 2e), with TENGi being able to detect low abundance signals such as $[\text{Glucose}+\text{K}]^+$, $[\text{18:1 LysoPC}+\text{K}]^+$, $[\text{Glucose}+\text{NH}_4]^+$, while DC nanoESI was not (Figure 2f and Figure S6).

Figure 2d-f. The behavior of four typical metabolite standards (adenine, lysine, glucose, and lysoPC) with TENGi and conventional DC nanoESI.

Figure S6. Comparison of TENGi and conventional DC nanoESI using four different common metabolites. Additional information of Figure 2d-f.

6. - The adduct formation used for metabolite table T1 is interesting, it suggests that TENGi produces adducts formation more so than traditional ESI/nanoESI. The author should comment on this aspect.

This is a very good point. In positive ion mode nanoESI, $[\text{M}+\text{H}]^+$, $[\text{M}+\text{Na}]^+$, and $[\text{M}+\text{K}]^+$ are very common adduct types, as shown in Figure 2d-f and Figure S6. $[\text{M}+\text{NH}_4]^+$ adducts were observed mainly because NH_4Ac was used during MSC processing, and the presence of NH_4Ac in solution will significantly promote $[\text{M}+\text{NH}_4]^+$ adduct formation. To further illustrate the effect of NH_4Ac addition, we have now added Figure S16 showing results that indicate that, indeed, the formation of ammonium adducts is related to the addition of ammonium acetate additives. When NH_4Ac was added, the major ion form for

glucose shifted from $[M+Na]^+$ to $[M+NH_4]^+$. We have also added a footnote to Table 1 discussing the points above.

Figure S16. TENGi and DC nanoESI MS results for a 10 μ M glucose solution with and without 5 mM NH_4Ac addition. Without NH_4Ac addition, glucose was mainly ionized as the $[M+Na]^+$ form with both TENGi and DC nanoESI; with the addition of NH_4Ac the major ion form switched to $[M+NH_4]^+$.

7. - Line 373-376 would lead the reader to expect that TENGi is untargeted, but with the raw data shown in the manuscript that isn't clear. TENGi is still as biased as ESI, however, it would be desirable to better understand to what degree the ions that are generated are more specific to the technique.

Apologies for the confusing sentence. We have now clarified it by changing it in the Discussion to “*Non-targeted metabolomics can be regarded as a hypothesis-generating process that holds great promise for discovering previously unknown biomarkers and biochemical pathway alterations*” and we added “*The lower sample volumes in TENGi MS make it an attractive option for rare sample metabolomics studies in either targeted or non-targeted mode*”.

TENGi belongs to the ESI family of ion sources, but with better performance. It can be used for non-targeted metabolomics experiments, as we show in this paper, or in targeted mode (by the use of a triple quad or Orbitrap in PRM mode). Both TENGi and ESI involve electric field-induced Taylor cone formation to generate charged droplets, Coulomb explosion and fission together with solvent evaporation to generate smaller droplets, ion evaporation/charge residue mechanisms to yield final gas-phase ions (*Anal. Chem.* 85, 2-9 (2013)). The major difference between TENGi and ESI is how the electric field is generated: conventional ESI uses a continuous DC power supply. TENG is a pulsed capacitor-like power supply that induces very strong electrostatic fields at the emitter tip. Because of the higher voltage created in the TENGi electrostatic field, low abundant species can be more effectively ionized and detected by MS. The newly-added data shown in Figure 3 illustrates this point further (as is also explained in our response to comment 4). TENGi in positive ion mode detects 300 more features than DC nanoESI (Figure 3d). In negative ion mode, the feature number detected by TENGi is almost twice that detected by DC nanoESI (Figure 3h).

8. - Typo in line 209 should indicate 3b and 3d (the correction).

Thank you, corrected.

9. - The authors should also avoid overstatements and be more precise including a comparison not only with cherry picked technologies that are clearly outperformed by theirs. Like in lines 263-274. There are several techniques which can work with way fewer cells and 1 million.

Thanks for the comment, you are correct. We have now removed that statement and added a more balanced discussion regarding the need for rare cell metabolomics methods, including references to a few successful cases in the literature: “*As MSCs are generally derived from patient donors, they are limited in supply, and costly to expand to sufficient numbers for metabolomics testing. This is a common issue with many cell types such as hematopoietic stem cells from blood, or tumor-derived stromal cells that are difficult to collect in large-enough numbers for metabolomics, leading to an increasing need for rare cell metabolomics methods. Li et al., for example, developed an isotope labeling nanoLC-MS method for small cell number untargeted metabolomics that has shown promise (50). Morrison et al. developed a targeted metabolomics method to screen about 50 different metabolites in haematopoietic stem cells and restricted haematopoietic progenitors (4).*”

10. On the biological insights side:

The work contains a large number of predictions of potential metabolite characteristics of a disease. However, beyond some retrospective analysis of literature around a handful of cherry-picked individual predictions, there is no real indication of the accuracy of any of these predictions, rendering the predictions of little real value it seems.

The main purpose of the two metabolomics applications presented is as a proof-of-concept to illustrate the potential of TENGi in rare sample metabolomics. We are aware of the limitations of the two studies, and have been very conservative when analyzing the data and presenting any biological conclusions. This is the reason why we implemented a very strict feature filtering strategy to limit the number of metabolites in the final dataset, focusing only in the most important effects. To make sure this is clear to the reader, we now have added extra statements to the end of the discussion of each of the two metabolomics studies to further emphasize this point: *“warrant further research in the future with larger cohorts and a suite of analytical platforms”*, *“The ability to detect the metabolomic changes with minimum sample amounts collected from the same subject after a few hours following a metabolic challenge brings a new analytical dimension where traditional approaches would typically not be readily applicable”* and *“With only 80,000 cells and <1 min detection time, TENGi sub-nanoliter metabolomics showed promising performance for small cell number metabolomics in these proof-of-principle studies. TENGi could likely benefit metabolomics studies on other rare cell types in their native state, rather than after many rounds of culturing. It not only can be applied to cases where relaxing sample volume requirements can facilitate both bio research and biomaterial manufacturing, like with MSCs, but is also highly suitable for cases where collecting large amount of sample is extremely difficult, as with EBC”*.

Reviewer #2 (Remarks to the Author):

General comments:

The research around “tiny metabolomics” is timely and important. The approach presented, more specifically the ionization method allowing sub-nanoliter sample consumption is novel and deserves publication. While 0.8 μL sample volume are filled in the emitter, and only pL are consumed for analysis using the TENGi nanoESI approach, a traditional DI nanoESI MS would require 20-30 μL sample volume per well, out of which 10-20 μL are consumed upon analysis. TENGi nanoESI requires changing emitters per sample. Thus, the huge advantage of the traditional “nano” method is sample throughput and automation. Despite this, I think that the new development will have its place as there is an increased interest in multi-omics studies having to deal with very precious limited sample volumes. The emerging topic of single cell MS based-omics can profit from the technology as well.

The authors show adequate proof-of-principle applications.

From the perspective of an analytical chemists, a more thorough comparison with DI-nanoESI MS and/or conventional metabolomics would have been desirable. The authors claim, that compared to traditional DI nanoESI MS, TENGi nanoESI has better signal-to-noise (S/N) ratios, but then this is shown only for one compound. It would have been desirable to proof with complementary conventional approaches that discoveries are not severely limited due to the sub-pL sample consumption or on the contrary even improved (as the authors claim). This would have been possible by simple experiments with standards spiked into biological matrices. In my opinion the authors fail to proof this specific point.

Thank you for your thorough comments and insight. We agree that in the previous version of the manuscript there was not sufficient data to sustain the claim that TENGi has better performance than nanoESI in terms of S/N. To this end, we have now added new data from new experiments using

metabolite standards and also more complex biological samples, as summarized in the table above. These data can better support our claims and show clear performance advantages of TENGi MS.

Overall, I recommend publication following addition of experiments, minor changes and clarifications.

Specific comments:

1. Line 40 “volume-limited biospecimens include cerebrospinal fluid”

The sample volume of CSF samples is in the mL range! Thus I do not regard this as a prime example of “tiny metabolomics”.

Thank you, we removed this statement. What we were trying to say is that CSF analysis would be doable without extracting as much sample from a patient.

2. The following statement (line 78-79) is not true: “However, conventional GC- or LC-MS-based cell metabolomics typically requires millions of cells.” Limits of detection of e.g. orbitrap technology using the conventional LC set up (2 mm column, flow rate 250-300 $\mu\text{L}/\text{min}$; injection volumes of μL) are in the pM to sub-nM range, which in turn enables to measure cell numbers of 10.000-100.000 cells on a routine basis.

The same holds true for the statement (line 280) “The analysis used only 80,000 cells per sample, which is more than an order of magnitude lower than that required for conventional LC-MS methods” Such a cell number could be analysed by conventional approaches. In fact, the concentration based LOD/LOQ are the same or even better in conventional LCMS (LOD of this study zeptomol/pL corresponding to nM).

Thanks for the thoughtful comments. We have removed the statement mentioned by the reviewer, and also modified a similar statement in the first paragraph of the Introduction that read “*Despite technological advances, many types of biospecimens still remain beyond the reach of LC-MS or GC-MS because they are difficult to collect in sufficient quantities to meet the sensitivity of such platforms.*” to “*Although state-of-the-art mass spectrometers can reach very low detection limits, the sample introduction techniques combined usually require certain volume of samples to work with. Therefore, many types of biospecimens remain beyond the reach of current metabolomics platforms for difficult-to-obtain biospecimens.*”

We have also changed a statements in the MSC metabolomics section that alluded to this point too. That section now reads: “As MSCs are generally derived from patient donors, they are limited in supply and costly to expand to sufficient numbers for metabolomics testing. This is a common issue with many cell types such as hematopoietic stem cells from blood, or tumor-derived stromal cells that are difficult to collect in large-enough numbers for metabolomics, leading to an increasing need for rare cell metabolomics methods. Li et al., for example, developed an isotope labeling nanoLC-MS method for small cell number untargeted metabolomics that has shown promise (50). Morrison et al. developed a targeted metabolomics method to screen 50 different metabolites in haematopoietic stem cells and restricted haematopoietic progenitors (4).”

3. Figure 1: Could you please specify how many scans are recorded per pulse? What is the RSD of the lactic acid signal obtained from multiple pulses? Is there an estimate of the lactic acid concentration range? What is the RSD of the pulse width?

With the Orbitrap settings used to perform the experiments shown in the paper, 6-7 scans were recorded per TENGi pulse. The pulse width RSD was 8.9 % and the RSD of the abundance of the lactic acid signal was 18.8%. This information is shown in Figure S9 (see below).

As suggested, we also performed additional experiments to quantitate lactic acid. EBC was collected from two healthy volunteers, concentrated 20-fold and analyzed by TENGi. These results are now provided in Figure S8.

Figure S9. Single TENGi pulse characterization. a) TIC of TENGi testing of a concentrated EBC sample from a healthy volunteer. b) the number of scans acquired for one TENGi pulse; c) the RSD of pulse width is 8.9%, and d) the RSD of the extracted ion chromatogram peak area for lactic acid is 18.8%.

Figure S8. Quantification of lactic acid in EBC. A healthy volunteer’s EBC was collected, concentrated 20-fold and analyzed by TENGi on an Orbitrap QE mass spectrometer. The lactic acid concentration in concentrated EBC was found to be 9.3-12.6 µM (marked blue on the plot, 0.46-0.63 µM in EBC before concentration).

4. In the introduction, the authors state that the combination of pulsed MS measurements and pulsed ionization allows a 100% sample utilization, which seems indeed an intriguing idea. Did I understand this correctly that a “real synchronization” is a future aim and for now the cycle times are independently controlled, but adjusted to fit? The authors do not further comment on how many measurement cycles are performed per pulse. This information is hidden in the method part.

Thank you for asking about this point. The reviewer is correct. To better explain this point, we have now rewritten the statement in the introduction to read “conventional ESI/nanoESI DI MS rarely achieves 100% duty cycle when coupled with pulsed mass analyzers. The result is unnecessary sample waste. Pulsed ion sources are more compatible with pulsed mass analyzers, such as Orbitrap or time-of-flight, and make the most of precious samples”. We have also added a new figure (Figure 1g) and corresponding discussion in the text, as follows: “An additional advantage is that TENGi’s pulsed characteristics are a better match to the intermittent nature of pulsed mass analyzer such as Orbitrap and time-of-flight. As is demonstrated in Figure 1g, conventional DC ESI/nanoESI supplies sample continuously, but only a fraction is utilized by the mass analyzer. A large fraction of the sample is wasted when the analyzer is busy scanning ions. In the case of TENGi, by synchronizing the ionization and mass analyzer cycles, each spray event can be more fully utilized to generate MS information, thus largely minimizing sample waste during analysis.”

Figure 1g). Illustration of sample supply and utilization in conventional DC ESI/nanoESI and TENGi. Because the mass spectrometer intermittently traps and scans ions, TENGi can be synchronized with these events so as to achieve higher sample utilization.

5. Table 1: please give significant numbers for fold change!!!

The significant figures for the fold change numbers have been corrected.

6. 1) Line 364..."With only 80,000 cells and <1 min detection time,I think this statement that the novel approach is superior compared to conventional metabolomics, should be rephrased, since it is not true in such a generalized manner. 2) The new technology is absolutely superior in terms of sample consumption, but the study does not allow to draw this final conclusion of higher sensitivity and thus higher discovery potential. This would have required the validation of the method by conventional metabolomics. The authors come to this conclusion, simply, because key findings regarding in the MSC study have not been published before. This is remarkable but not a stringent argument with regard to detection power. There might be multiple reasons why previous studies failed to discover certain metabolic rearrangements.

1) We have changed the corresponding statement to "TENGi sub-nanoliter metabolomics showed promising performance for small cell number metabolomics in proof-of-principle studies. TENGi could likely benefit metabolomics studies of other rare cell types without the need for many rounds of culture."

2) We agree with the reviewer and have made several modifications to the manuscript to address this point, which was also brought up by reviewer #1 (please see responses above). We have now performed experiments with a very dilute liver lipid extract ($25 \mu\text{g mL}^{-1}$), and added the results as Figure 3. Overall, TENGi yielded much higher abundances than conventional DC nanoESI. The number of features detected by TENGi in positive and negative ion modes was therefore much higher than that detected by DC nanoESI. Detailed discussions about this point have also been added in the Result section in the revised manuscript.

7. All measurements have been carried out in negative mode. Is this a technological restriction of the ionization? Would pos/neg switching be possible? Please comment!

This is a very good point. TENG-MS technology is not restricted to one ionization mode, as both positive and negative charges are generated during the operation cycle. EBC metabolomics experiments were carried out in negative mode as in previous work we learned that this gave us the most biological information (J. Prot. Res., 2020, 19(1), 144-152). MSC experiments were done in positive mode, as this is our default for discovery experiments. Experiments with liver extract further demonstrate that TENGi can be carried out in both modes (Figure 3). However, to further illustrate this capability, we have now added polarity switching experiments of liver lipid extract as Figure 9. Both negative and positive spectra were collected with only sub-microliter sample loading and sub-nanoliter sample consumption.

Figure 9. Polarity switching TENGi MS to obtain both positive and negative signals in the same experiment with sub-nanoliter sample consumption. The sample tested was a $50 \mu\text{g mL}^{-1}$ liver lipid extract. a) TIC of TENGi coupled with polarity switching, Negative and positive signals are detected alternatively. b) Detail of emitter with $0.8 \mu\text{L}$ sample loaded. c) mass spectra for negative pulses; d) mass spectra for positive pulses.

8. 1) Line 422-424 “Twenty (for MSC experiments) to forty (for EBC experiments) TENGi pulses were

obtained for each sample. Three (for MSC experiments) or five (for EBC experiments) of such pulses were selected based on the extracted ion trace pulse areas for the IS (1 μ M 13 C-tyrosine), discarding any outliers to ensure maximum reproducibility”

This is my major concern, where clarification is needed! Have all data (the nice RSD values shown for QC samples and internal standards) been based on such a selection, 3-4 pulses out of 20/40? This is a very problematic data evaluation, where the statistical foundation of this outlier selection (the majority of pulses are outliers?) becomes very fragile. Please comment! This type of data selection makes the results rather arbitrary!

We apologize for the misleading statement. What we meant is that we randomly selected pulses for obtaining mass spectra for metabolomics fingerprinting. To make it crystal clear, we have now added a new Figure in the supporting information section (Figure S21, also shown below), to better illustrate the process and explain the rationale for outlier identification. We have also edited the statement quoted by the reviewer to *“Outlier pulses were discarded based on the extracted ion trace pulse areas for the IS, and three (for MSC experiments) or five (for EBC experiments) of the remaining pulses were randomly selected to generate the representative spectrum for the corresponding sample. Outlier recognition was based on a $\pm 20\%$ interval centered around the median area of the IS EIC. More details are provided in Figure S21, including the statistical basis for the outlier identification process.”* As the reviewer notes, outlier identification requires robust statistics, that’s exactly why we collected 20-40 pulses: to increase the number of technical replicates to help identify and avoid extreme pulses (outliers). The reason why we only used 3-5 randomly selected pulses, and not all the pulses, is because we wanted to establish that even a small number of pulses suffices for generating robust metabolomics data for rare samples.

Figure S21. Internal standard (IS)-assisted outlier pulse identification. a) TIC for an EBC sample. b) EIC for the ^{13}C tyrosine IS. c) After obtaining the median value for the IS EIC pulse areas, the $\pm 20\%$ interval was calculated. Pulses with IS area below or above this range were deemed outliers. They are marked with stars in b).

What is the RSD of the 40 pulses?

The pulse area abundance variation data have been added to Figure S21a. The RSD was 17.8%.

Reviewer #3 (Remarks to the Author):

The authors presented a new mass spectrometry the triboelectric nanogenerator inductive

nanoelectrospray ionization with making full use of each spray event by consuming only sub-microliters of sample. Different with the previous work, this manuscript focused on converging this device with exhaled breath condensates and showed the advancement in easy sample collecting. This manuscript is acceptable to Nature Communications after some modification. The detailed comments are as follows:

1. Even though the detailed part related to TENG is dealt in previous work (ref. 7), TENG part should be added to the manuscript. (e.g. fabrication process, simple principle, and basic electrical output)

Thank you for this comment. To address it, we have added more detailed illustrations and device details to the revised manuscript. These include:

1) An illustration of the TENG device fabrication (Figure 1b). The type of TENG utilized here is a sliding freestanding (SF) TENG (Figure 1b). The TENG device is constructed with one pair of triboelectric layers and one pair of copper film electrodes. The triboelectric layers were made of a nylon sliding layer (12x12 cm) and fluorinated ethylene propylene (FEP) stationary layer (24x12 cm).

Figure 1b. Schematic illustration of the SF TENG device used in this study.

2) We have also added a figure describing the TENG electrification mechanism, as Figure S2.

Figure S2. TENG charging mechanism illustration. a) With no electrode movement, no charge is generated; b) Sliding the top electrode to the right results in a negative voltage output to the ion source; c) Electrode movement to the left results in a positive voltage output to the ion source..

3) The electrical characterization results for the TENG device have been rearranged from Figure S1a-c to Figure 2a-c, to give them more prominence and bring the reader's attention to them.

2. The order of the figures is confusing due to the not matching with the order of the result part. The readability would be increased with a suitable relocation.

Apologies, we have now rearranged the figures and also added in a number of new figures to both the main text and the Supporting Information. Detailed information about all these changes is summarized in two tables placed at the very beginning of this document.

3. In Figure 1c and S3, the meaning of values on graph and more detailed explanation are required for easy understanding to readers.

Figure 1c has been rearranged to Figure 1f in the revised manuscript (Figure S3 is still Figure S3). We have added additional notes to the Figures and the associated legends to make the illustrations clearer to the readers.

4. In Figure S4, the authors showed the preserved samples with 10 days. However, how can we know the preserved samples show the same characteristic compared with 0 day sample?

Thanks for the suggestion. Mass spectra obtained for the sample after 10 days of preservation have now been added to Figure S4 as Figure S4e and S4f. The distribution of the major peaks is very similar to the spectral abundances of the fresh sample (Figure S3a and 3b). In cases when sample stock is extremely small, this method could be used for saving the sample for additional metabolite annotation experiments *etc.*

Figure S4e-f). Blank-subtracted mass spectra in both positive and negative ion modes after 10 days preservation. The relative abundances of the most abundant peaks are similar to those obtained for the fresh samples, shown in Figure S3a and 3b.

5. In the part of Figure S6b, the elucidation of how to obtain the sensitivity and specificity needs to be supplemented.

Thank you for the suggestion. For this figure (now renumbered to Figure S11 in the revised manuscript), we have now clarified how sensitivity and specificity were calculated in the caption, as follows: *“Sensitivity is defined as the proportion of actual positives that are correctly identified (also known as the true positive rate). In this case, it is calculated as the number of true “Post” positives samples correctly identified (n=8) divided by the total number of “Post” samples (n=9). Specificity is the proportion of actual negatives that are correctly identified (also known as the true negative rate). Here it is calculated as the total number of “Pre” samples that are correctly identified (n=7) divided by the total number of “Pre” samples (n=10).”*

6. There is no Figure 4b. It needs to be added to the manuscript in appropriate part.

Corrected. Original Figure 4 has been renumbered to Figure 6 in the revised manuscript.

7. In the starting part of the sentence dealing with Table S2, there is a typographic error and this punctuation needs to be modified.

Corrected.

REVIEWERS' COMMENTS

Reviewer #1 (Remarks to the Author):

The authors did a very good job in addressing most of my concerns. I still find the analysis of results from CF patients too speculative. It is not just a matter of increasing the cohort size, but mostly the interpretation of the metabolic difference. IS there any sort of positive control?

Reviewer #2 (Remarks to the Author):

The authors revised the manuscript carefully, addressed all critical points and open questions and did their best to reveal technical details on the method together with a convincing application story. Thus, I support publication in its current form.

REVIEWER COMMENTS

Reviewer #1 (Remarks to the Author):

The authors did a very good job in addressing most of my concerns. I still find the analysis of results from CF patients too speculative. It is not just a matter of increasing the cohort size, but mostly the interpretation of the metabolic difference. IS there any sort of positive control?

We are grateful that the reviewer has found our revisions adequate, and we are thankful for the feedback, it has certainly made this a better manuscript. We do share the sentiment that the CF work is, to some extent, speculative. We view the CF work presented here as a pilot study to demonstrate the capabilities of the TENGi MS technique, rather than a full-scale study to understand the metabolome changes induced by glucose challenge in CF. As the reviewer mentions, the cohort size would have to be increased significantly to reach any clinically-relevant conclusions. The reviewer also discusses the possibility of adding a positive control, but this is not possible in the current study design, unfortunately. Please rest assured that this is an ongoing research topic in collaboration with our colleagues at Emory University, and longer-term longitudinal studies are currently underway.

Reviewer #2 (Remarks to the Author):

The authors revised the manuscript carefully, addressed all critical points and open questions and did their best to reveal technical details on the method together with a convincing application story. Thus, I support publication in its current form.

We are glad to hear that this reviewer is satisfied with our response. Thank you again for all the effort put in the peer review process, we feel the manuscript is much better now than it was initially.